# Model-simulated hydroclimate in the East Asian summer monsoon region during past and future climate: a pilot study with a moisture source perspective

**Astrid Fremme**[1,a]**, Paul J. Hezel**[1]**, Øyvind Seland**[2]**, and Harald Sodemann**[1]

[1]Geophysical Institute, University of Bergen, and Bjerknes Centre for Climate Research, Bergen, Norway
[2]Norwegian Meteorological Institute, Oslo, Norway
[a]now at: Statkraft, Oslo TS1, Norway

**Correspondence:** Harald Sodemann (harald.sodemann@uib.no)

**Abstract.** Here we present a pilot study of the sensitivity of summer monsoon precipitation in the Yangtze River Valley (YRV; 110–122° E and 27–33° N, eastern China) to climatic boundary conditions from the Last Glacial Maximum (LGM), pre-industrial conditions, and the Representative Concentration Pathway 6 emission scenario from two different climate models. Using a quantitative Lagrangian moisture source diagnostic based on backward trajectories, we are able to interpret changes in precipitation amount and seasonality in terms of processes at the source regions and during transport that contribute to YRV precipitation. Thereby, we gain insight into influential processes and characteristics related to precipitation variability and the sensitivity of the summer monsoon hydroclimate in East Asia to boundary-condition changes in models. Comparing 10-year time slices similar to present-day conditions from the NorESM1-M and CAM5.1 models to reanalysis data reveals overall similar moisture source regions, albeit with a tendency for a more local precipitation origin in the climate models. The general characteristics of the moisture sources and moisture transport in the YRV are relatively stable across different climate forcings, both concerning the mean location of source regions, transport distance, and the relative contributions of moisture from land and ocean areas. Changes regarding regional precipitation contributions from the East Asian continent indicate that precipitation recycling responds to different climate forcings. We interpret these findings such that models to first order respond with a scaling rather than reorganisation of the hydroclimate to climatic forcing, while land–atmosphere interactions play an important, but secondary, role. If the model simulations are accurate, the moisture source regions and thus the general processes of precipitation in the YRV could remain relatively stable across different climates. However, some differences in moisture source conditions are larger between the different climate models than between different climatic boundary conditions in the same model. It may therefore be possible that current climate models underestimate the potential for non-linear responses to changing boundary conditions, for example due to precipitation recycling. Although limited by the relatively short analysis period, our findings demonstrate that the diagnosis of moisture sources provides a useful additional perspective for understanding and quantifying precipitation mechanisms and the hydroclimate simulated by models and enables more detailed evaluation of model simulations, for example using paleoclimate records.

## 1 Introduction

Climate variations on inter-annual to millennial timescales are intimately linked to hydroclimate variability. Hydroclimate variability in the East Asian summer monsoon (EASM) region is of particular relevance, as changes in this region can have important consequences for other parts of the Earth's climate. Furthermore, the livelihood of a large population is adapted to the present climatic conditions in this region (Zong and Chen, 2000). Paleoclimate archives in East Asia

have been pivotal for the general understanding of monsoon systems and their variability over long timescales (Thompson et al., 2000; Wang et al., 2001; Dykoski et al., 2005; Hu et al., 2008; Wang et al., 2017). For example, stable isotope parameters, such as $\delta^{18}O$ in carbonates, ice cores, and tree rings, are commonly interpreted as monsoon strength or monsoon precipitation intensity, thus reflecting regional precipitation amount (Wang et al., 2001).

However, some studies show that the isotopic information stored in these records may at times be influenced or even dominated by other effects, such as circulation-induced moisture source changes (Maher and Thompson, 2012). Changes in land-surface parameters are a further factor that has not yet been thoroughly explored but could potentially play an important role (Fremme and Sodemann, 2019). Adopting a moisture source perspective has been shown to be potentially valuable for interpreting the paleoclimate information contained in stable water isotopes from different archives in the East Asian monsoon region using reanalysis data (Liu et al., 2014; Baker et al., 2015). Insight into model-simulated hydroclimate variability could thus be highly beneficial with regard to moisture source and transport changes to interpret paleoclimate records and CE1 provide ground truth for model-simulated present-day hydroclimate, as well as enable more reliable future climate projections. However, the complex interaction of land and ocean, the orography, and the atmospheric dynamics in this region render the identification of mechanisms that underlie precipitation changes challenging. Here, we apply a robust diagnostic for precipitation sources and transport based on backward trajectories, which has so far only been used with reanalysis data, to general circulation model (GCM) output. Our aim is to understand which mechanisms impact simulated hydroclimate variability in the EASM region across different climatic conditions, specifically in the Yangtze River Valley (YRV).

Simulating the hydroclimate in the global monsoon regions has been notoriously challenging, both for past and present climates. In addition to variability in different timescales, climate models struggle to reproduce the spatial details of precipitation and other relevant variables (Braconnot et al., 2012). For example, a weaker meridional temperature gradient in the troposphere, arising from the differential heating over the Tibetan Himalayas and the Indian Ocean, leads in many models to a later onset and weaker monsoon circulation with less precipitation (Ashfaq et al., 2017). A major reason for model deficiencies clearly lies in the limited horizontal resolution in common GCMs, which requires a large share of processes to be handled by sub-grid-scale parameterisations. At these grid scales, the complex interplay of physical and dynamical factors is often represented poorly, including convection, low-level jets, orography, and land-surface processes (Webster et al., 1998; Hoyos and Webster, 2007; Seo et al., 2013). Hydroclimate variability may be particularly sensitive to such interplay of different factors, as the atmospheric water cycle is, for example, connected to the land and ocean surface by surface fluxes and precipitation, involving precipitation recycling (Fremme and Sodemann, 2019; Gimeno et al., 2021). Since water vapour is a central feedback mechanism of the climate system, better understanding of the interplay between different mechanisms in a model's hydroclimate also benefits further GCM development.

Small-scale variability of precipitation in space and time, both in observations and simulations, renders precipitation a particularly challenging variable for studying processes contributing to hydroclimate variability. While precipitation is a key target variable of climate models, its representation in the grid-scale microphysics and in moist convection parameterisations differs markedly between models. In this context, more robust means for hydroclimate evaluation than simulated precipitation can be a valuable asset in evaluation studies. For example, the horizontal moisture flux, expressed as integrated vapour transport, has been shown to more reliably predict extreme precipitation than simulated precipitation itself (Lavers and Villarini, 2013). Horizontal moisture flux and integrated vapour transport can effectively map moisture transport. However, the evaporation sources corresponding to precipitation, often referred to as the moisture sources, are most readily obtained from Lagrangian backward trajectory calculations (Stohl et al., 2008; Sodemann et al., 2008; Bohlinger et al., 2017; Fremme and Sodemann, 2019). We hypothesise that combining diagnosed moisture sources from Lagrangian methods with model precipitation allows us to identify causes for hydroclimatic variability more readily than precipitation alone.

Previous studies that investigated the contribution of land and ocean areas as moisture sources to the EASM region from reanalyses differ markedly in their results. The most important oceanic source regions appear to be the Arabian Sea, the Bay of Bengal, the South China Sea, and the western Pacific (Wei et al., 2012; Wang et al., 2018), with southwesterly moisture transport providing a large fraction of the water vapour for the East Asian monsoon. While some authors emphasise the importance of oceanic regions over land areas (Zhou and Yu, 2005; Chen et al., 2013), several authors have also determined that land areas contribute substantial amounts (Wei et al., 2012; Sun and Wang, 2015). It appears that quantification of contributions from different source areas is strongly influenced by the respective methods. In a study based on ERA-Interim reanalysis data, Fremme and Sodemann (2019) analysed the processes leading to seasonal and inter-annual variability of the YRV precipitation variability using a Lagrangian moisture source and transport diagnostic that determines source regions without the need to pre-specify the atmospheric lifetime of water vapour (Sodemann, 2020). Based on the quantification of moisture sources for the period 1980–2016, the study of Fremme and Sodemann (2019) revealed a major role of land-surface processes, leading to several cycles of precipitation recycling for 50 %– 65 % of the rainfall in the YRV. Since previous studies map-

ping the moisture sources for precipitation in eastern China only covered present-day periods from reanalyses, moisture source changes across different climates, and in particular the respective role of land contributions, have so far not been assessed with such a method.

Here we use the Lagrangian moisture source diagnostic of Sodemann et al. (2008) to obtain the moisture sources of the YRV as a core region of the East Asian monsoon system using simulations from two climate models for different climatic periods. In this pilot study, we use the moisture diagnostic for the first time with free-running model simulations, thereby avoiding the influence from data assimilation present in reanalysis data. To this end, we first assess how different climate models transport moisture to the YRV during the monsoon season in a present-day climate using results obtained previously from reanalyses as a reference (Fremme and Sodemann, 2019). From simulations with different climatic boundary conditions, we then identify how models represent hydroclimate variability during the simulated East Asian summer monsoon to orbital forcing and ice-sheet topography from analysing time slices of an uncoupled simulation of the Last Glacial Maximum (LGM). Furthermore, the changes of the monsoon system in a future climate scenario with increased atmospheric $CO_2$ concentrations are assessed in a coupled model run for a time slice near the end of the 21st century. Based on our findings, we then discuss in particular the role of land-surface processes and conclude with remarks on the potential of a moisture source perspective for understanding hydroclimate variability and for interpreting paleoclimate records from the East Asian monsoon region and future model studies.

## 2 Methods and data

The aim of our study is to investigate the response of the hydroclimate to different climate model configurations for the East Asian monsoon from a moisture source perspective. As in the study of Fremme and Sodemann (2019), we use the YRV as a focus region, defined here as the lower reaches of the Yangtze River at 110–122° E and 27–33° N, eastern China. We apply the widely used quantitative Lagrangian moisture source and transport diagnostic by Sodemann et al. (2008) based on FLEXPART (FLEXible PARTicle transport model; Stohl et al., 2005; Pisso et al., 2019) backward air parcel trajectories to find the moisture sources of the YRV in different climate model simulations and time slices. We first describe the setup of the climate model simulations, followed by an explanation of the trajectory calculation setup and a description of the moisture source diagnostics, with the parameter choices for the climate model data.

### 2.1 Climate model simulations

In total we analyse moisture transport and sources for the YRV in four climate model simulations, contributed by two different climate models (Fig. 1a, Table 1). The two models are the atmosphere-only Community Atmosphere Model CAM5.1 (Neale et al., 2012) and the fully coupled ocean–atmosphere Norwegian Earth System Model (NorESM1-M; Bentsen et al., 2013). For each model, we analyse a control simulation for present climate conditions to assess how precipitation is represented in comparison to reanalysis data and one simulation of a different climate (Fig. 1a). For the uncoupled, atmosphere-only simulations with CAM5.1, we analyse a simulation of the pre-industrial period (PIN) as a reference and then evaluate the sensitivity by comparing to a simulation of the Last Glacial Maximum (LGM) climate. For the coupled ocean–atmosphere simulations with NorESM1-M, a time slice from a control run with present-day conditions (CTL) is used as a reference, and a time slice from a transient simulation with the CMIP5 Representative Concentration Pathway emission scenario reaching a radiative forcing of about $6 \, \text{W m}^{-2}$ by the end of the 21st century (RCP6) allows us to assess sensitivity to a future climate. The RCP6 scenario is an intermediate emission scenario of the CMIP5 tier 1 category, supplementing the core scenarios RCP4.5 and RCP8.5 (Taylor et al., 2012). We make use of this less commonly used emission scenario here, since a NorESM1-M simulation with 3-hourly output of three-dimensional model fields needed by the moisture source diagnostics was available for this study. Throughout the paper, we refer here to the emission scenario itself as RCP6, while we refer to the NorESM1-M simulation with the RCP6 emission scenario and corresponding results as "RCP".

By comparing moisture source results from reanalyses with the near-present-day simulations and the near-present-day with a different climate from the corresponding model, these four model simulations enable an assessment of how models represent moisture source changes for changing climate conditions (Fig. 1a, horizontal arrows). For the calculation of trajectories and diagnosis of moisture sources, it was necessary to output and archive three-dimensional model fields of wind, temperature, and humidity at a 3 h time interval (Cassiani et al., 2016). This requirement poses severe limitations on the duration and number of climate simulations that can be performed and archived over a longer time for such analyses. In this pilot study, we use 10-year time slices for the trajectory calculation and analysis.

The two atmosphere-only CAM5.1 simulations PIN and LGM were run with a resolution of $0.9 \times 1.25°$, with 30 vertical levels, using the finite-volume dynamical core (Neale et al., 2012). Both simulations were run for 30 years each, starting after a 3-year spin-up period (Table 1). The PIN simulation used climatological sea surface temperatures (SSTs) and sea ice from the merged Hadley–NOAA/OI sea surface temperature and sea ice concentration dataset (Hurrell et al.,

**Table 1.** Characteristics of the four climate model simulations with CAM5.1 and NorESM1-M and the ERA-Interim reference dataset.

| Name | Description | Model | Grid resolution | Levels | Configuration | Years analysed |
|------|-------------|-------|-----------------|-------:|---------------|----------------|
| LGM | Last Glacial Maximum | CAM5.1 | $1.25 \times 0.94$ | 30 | Prescribed ocean | 0010–0019* |
| PIN | Pre-industrial (control) | CAM5.1 | $1.25 \times 0.94$ | 30 | Prescribed ocean | 0010–0019* |
| CTL | Historical control | NorESM1-M | $2.50 \times 1.88$ | 26 | Fully coupled | 2001–2010 |
| RCP | IPCC RCP6 scenario | NorESM1-M | $2.50 \times 1.88$ | 26 | Fully coupled | 2061–2070 |
| ERAI | ERA-Interim reference | IFS | T255 (interpolated to $1.0 \times 1.0$) | 61 | Prescribed ocean | 1980–2016 |

\* Model years.

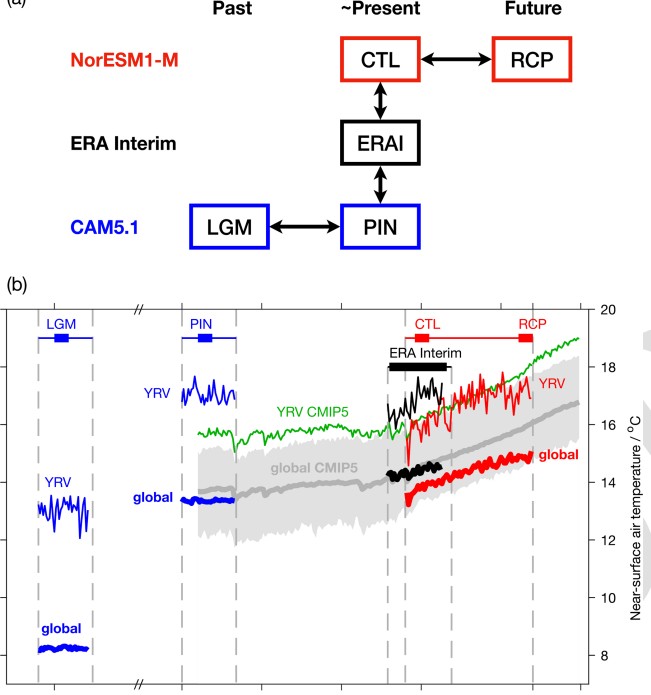

**Figure 1.** Overview of the study setup. **(a)** Schematic for the comparison between the simulations with NorESM1-M (red) for a near-present (control, CTL) and a future climate (Representative Concentration Pathway 6, RCP), with CAM5.1 (blue) for a near-present (pre-industrial, PIN) and a past climate (Last Glacial Maximum, LGM), and the reference present-day climate from ERA-Interim (ERAI, black). **(b)** Compilation of surface air temperatures from all datasets for global mean (thick lines) and the Yangtze River Valley (thin lines). Global mean and spread of surface air temperatures from the CMIP5 multi-model mean for RCP6 from 1860 to 2075 (Taylor et al., 2012) are shown in grey, and the green line shows the mean for the YRV domain. The duration of simulations with CAM5.1 and NorESM1-M is shown with thin blue and red lines on top. Thick blue lines show the years for which moisture sources have been analysed. Note the broken time axis between 1850 and 21 000 years ago (21 kyr).

2013), averaged for the period of 1870–1899. Atmospheric carbon dioxide was prescribed at a mixing ratio 284.7 ppmv and atmospheric methane at a mixing ratio of 791.6 ppbv.

The LGM simulation used the topography and ice sheets specified as in PMIP3 (Braconnot et al., 2012; Abe-Ouchi et al., 2015) and CMIP5 (Taylor et al., 2012) experiments for 21 kyr. LGM SSTs and sea ice climatology were obtained from 21 kyr simulations performed at the National Center for Atmospheric Research (NCAR, Boulder, Colorado, USA) for the LGM (https://www.earthsystemgrid.org/dataset/ucar. cgd.ccsm4.b40.lgm21ka.1deg.003M.html, last access: 1 January 2019 TS2). Sea level was kept at PIN conditions during the LGM simulations, potentially introducing unrealistic land–atmosphere interaction in the region of the Maritime Continent and western Pacific Warm Pool region. Mixing ratios of atmospheric greenhouse gases were 185 ppmv for $CO_2$ and 185 ppmv TS3 for methane during the LGM simulation. In the YRV region, temperatures are simulated to be on average 13.0 °C, which is 4.8 K warmer than the global average of 8.2 °C (Fig. 1b, thick and thin blue lines). This global average LGM temperature change is larger than estimates from reconstructions (4.0 K, Annan and Hargreaves, 2013). During PIN, in comparison, the global mean surface air temperature is 13.4 °C, which is 3.6 K colder than in the YRV (17.0 °C). The changes between LGM and PIN detected here for surface air temperature and precipitation are comparable to those in the PMIP3 simulations conducted as part of CMIP5 (Harrison et al., 2014). We also note an increase in zonal wind speed of about 1 m s⁻¹ across a broad band from northern India to the Philippines in the LGM (Appendix A).

The coupled NorESM1-M simulations CTL and RCP had a horizontal resolution of $1.88° \times 2.50°$ with 26 vertical levels and were run as a continuous simulation for an 80-year period from 1990–2070 (Table 1). NorESM1-M is based on CCSM version 4 (Gent et al., 2011), with the atmospheric component being CAM4-Oslo (Kirkevåg et al., 2013) and the CLM4 land component (Lawrence et al., 2011). In these simulations, greenhouse gases ($CO_2$, $CH_4$, $N_2O$, CFC-11, and CFC-12), volcanic $SO_4$, total solar irradiance, and the ozone distribution were prescribed for the historical period (Lamarque et al., 2010) and simulated from 2005 onwards (Stohl et al., 2015). The climate simulations with NorESM1-M are separated into a control simulation (CTL) for present day and a future climate simulation (RCP). For the observational period, the CTL model simulations are substantially colder than ERA-Interim (14.4 °C globally and 16.8 °C in the YRV). The

global average surface air temperature between CTL and RCP increases by 0.7 K for the RCP6 scenario, from 13.7 to 14.5 °C. NorESM1-M is thereby near the lower end of the range of climate model simulations that contributed to CMIP5 (Fig. 1b, grey shading). In the YRV region, the temperature difference between CTL and RCP is 1.1 K. Notably, during CTL, the YRV region is only 2.1 K warmer than the global average and 2.4 K during RCP, in close agreement with the CMIP5 mean (Fig. 1b, green line).

A general cold bias over land in NorESM1-M was documented earlier (Seland et al., 2020, their Fig. 14). While the ultimate reason remains unknown, this cold bias could be related to aerosol properties. Clearly, some differences between simulations and reanalyses are always expected, as illustrated by the spread of the CMIP5 models, which may be due to differences in the boundary conditions, model resolution, atmospheric dynamics, physics parameterisations, and ocean model. We will return to some of the differences noted here when discussing the moisture source results.

## 2.2 Setup for moisture source analysis

The moisture sources for each climate model simulation were identified using the Lagrangian moisture source diagnostic WaterSip (Sodemann et al., 2008). The diagnostic identifies the evaporation sources and transport pathways of precipitation falling in a target domain from specific humidity changes along backward trajectories. A particular advantage of this offline method is that it can be applied to meteorological fields and trajectories from different sources. Here, we calculated a large number of air parcel backward trajectories using FLEXPART–NorESM/CAM (V1) (Cassiani et al., 2016) for the 3-hourly model level output from the climate models NorESM1-M and CAM5.1. This pilot study is the first use of the moisture source diagnostic with climate model output data.

By means of the so-called domain-filling mode, FLEXPART constantly released new particles in proportion to the mass flux into the domain. For the FLEXPART setup, the initial 50 000 air parcels of equal mass ($\sim 4.54 \times 10^{11}$ kg per parcel) in PIN and LGM and 25 000 air parcels of equal mass ($\sim 1.15 \times 10^{12}$ kg per parcel) in CTL and RCP were initiated in the atmosphere over the YRV region (108–124° E and 25–35° N). Using 3-hourly, three-dimensional wind fields from the climate models, FLEXPART–NorESM/CAM then calculated air parcel movements backward for at least 16 d, starting from the YRV domain defined above. A sensitivity study revealed only very minor differences between the results using 25 000 or 50 000 air parcel trajectories (not shown). Running for one time slice at a time, the FLEXPART model then stored air parcel trajectories including their horizontal and vertical position, air temperature, specific humidity, and surface characteristics at the position of each air parcel at a 3 h time interval for processing with the moisture source diagnostic WaterSip (Sect. 2.3).

Due to the substantial computational requirements regarding the post-processing for moisture source identification and storage constraints, the trajectory calculations covered 10-year time slices for each climate model simulation (Table 1). For PIN and LGM, the 10-year periods were chosen such that the YRV precipitation mean was similar to that of the full simulation period. Due to the climatological SST forcing, the PIN and LGM simulations are not affected by ocean-induced inter-annual variability. The time slice from both the PIN and LGM simulations thus covered the model simulation years 0010–0019. For CTL, the latest 10-year period overlapping with the ERA-Interim period was chosen (1996–2005). Since this included the strong El Niño/La Niña event of 1997/1998, we compared the results with and without including the years 1997 and 1998 in the analysis. If was found that the differences with and without the years 1997/1998 were substantially smaller than the MJJ mean standard deviation. As the analysis leads to the same results in both cases, we decided to include the years 1997/1998. For the RCP time slice, we chose the 10-year period at the end of the future climate simulation (2061–2070). Nonetheless, the reliability of our results from this pilot study is limited due to the relatively short analysis periods, in particular with regard to impacts from climate variability. Future climatological studies should aim to perform the moisture source diagnostics over the conventional duration of 30 years.

## 2.3 Parameters for moisture source diagnostics

Next, the moisture source diagnostic WaterSip was used to evaluate the trajectories corresponding to each individual precipitation event in the YRV domain (for technical reasons here expanded to 110–122° E and 27–33° N). Evaluating each air parcel trajectory backward in time, specific humidity changes along the way provide an estimate of either a contribution of water vapour from surface evaporation to the air parcel or the loss of water vapour due to precipitation along the way (Sodemann et al., 2008). Thereby, either evaporation or precipitation is assumed to dominate within a given time interval. Importantly, the contribution of surface evaporation at each moisture source is quantified relative to the water vapour already contained within an air parcel at a given time. Assuming well-mixed conditions within the air parcel (but not within the column), all water vapour in the air parcel contributes to precipitation according to the relative share of the moisture sources. Finally, the contribution of each individual source to total precipitation in the YRV is found from the precipitation-amount-weighted integral of all trajectories within a given time interval. In addition to the sources' locations, characteristics of the moisture sources and transport are identified, such as the moisture source distance, temperature, or surface type (for further details see Sodemann et al., 2008; Fremme and Sodemann, 2019). A particular advantage of the WaterSip method is that there is

no need to pre-specify the lifetime of water vapour (Sodemann, 2020; Gimeno et al., 2021).

The WaterSip diagnostic has here been adapted to the target region and climate model input data by evaluating and adjusting several threshold parameters. Sensitivity to different time-step lengths and thresholds for specific humidity changes ($\Delta q_c$) and relative humidity ($RH_c$) was assessed in detail for a subset of the data (Appendix B). Based on these sensitivity tests, the threshold for identifying significant changes in specific humidity per time step ($\Delta q_c$) was set to 0.1 g kg$^{-1}$ at a 6 h$^{-1}$ time step. Precipitating trajectories in the target region were identified from a relative humidity threshold ($RH_c > 80\%$).

We diagnosed the moisture transport for 15 d along each trajectory. No distinction has been made between moisture sources identified in the boundary layer or within the free troposphere for this study. On average, the combination of parameter choices allows us to assign moisture sources to about 95 %–98 % of the precipitation estimated by the WaterSip diagnostic. This percentage is larger than usually obtained with trajectories from reanalysis data, possibly because there are no inconsistencies introduced from data assimilation in the free-running climate simulations.

## 2.4 Reference analysis

Since each climate model may have its own model-specific representation of the hydrological cycle, we evaluate their performance using data from the European Centre for Medium-Range Weather Forecast's (ECMWF's) ERA-Interim reanalysis (Dee et al., 2011) as a reference (Table 1). Several studies have shown that the ERA-Interim reanalysis provides a realistic representation of the climatic conditions in the YRV region (Lin et al., 2014; Huang et al., 2016).

We furthermore compare moisture sources identified from the climate models to the moisture sources from reanalysis data. Fremme and Sodemann (2019) analysed YRV moisture sources based on ERA-Interim data and the WaterSip diagnostic from a global trajectory dataset (Läderach and Sodemann, 2016), spanning the period 1980–2016 (ERAI; Table 1). The WaterSip parameters for the reanalysis climatology were the same as used here for the climate model simulations. In the reference analysis, 95 % of the estimated precipitation could be attributed to corresponding moisture sources.

## 3 Results

Now, we first evaluate the performance of the two climate models in representing East Asian monsoon precipitation. To this end, the near-present control simulations of CTL and PIN are compared to the reference climatology based on the ERA-Interim reanalysis (Fig. 1a, vertical column and arrows). Then, precipitation in the paleo-simulation (LGM)

and in the future scenario simulation (RCP) is compared to precipitation from the corresponding control simulation from the same climate model.

## 3.1 Summertime YRV precipitation in the near-present simulations PIN and CTL

In our analysis, we focus on the YRV summer monsoon precipitation, which peaks during May, June, and July (MJJ). According to the ERA-Interim dataset, summer precipitation is at a maximum in a broad region between the Bay of Bengal (BoB) and the southern border of the Tibetan Plateau (Fig. 2a). A distinct orographic precipitation gradient is apparent along the western edge of the Indian Peninsula. A further precipitation maximum is apparent over the Philippines and the western Pacific as part of the northward-displaced Intertropical Convergence Zone (ITCZ) In the YRV region (Fig. 2a, red box), precipitation shows a relatively weak north–south gradient with average values of 5–9 mm d$^{-1}$ and with the highest values south and west within the region.

In comparison, the CTL simulation exhibits a similar range of precipitation values (Fig. 2b). The maximum along the southern edge of the Tibetan Plateau and western India is more spread out and reaches well above 12 mm d$^{-1}$. However, there is a clear lack of precipitation over the BoB and the Philippines. Precipitation underestimation dominates in the CTL simulation (Fig. 2c, red shading), with an overall bias of $-1.0$ mm d$^{-1}$ for the displayed EASM domain and $-1.4$ mm d$^{-1}$ in the YRV (Table 2). Overestimation of precipitation ($> 6$ mm d$^{-1}$) is apparent at the southern edge of the Tibetan Plateau and to the west of the Tibetan Plateau. In the YRV, MJJ precipitation is between 0–3 mm d$^{-1}$ lower than ERA-Interim. Here, the CTL simulation has a precipitation range of 4–6 mm d$^{-1}$, clearly missing the finer details of spatial variability. Despite such local biases, we at first order consider the CTL simulation to reasonably reproduce most of the large-scale features of summer precipitation in East Asia when compared to the ERA-Interim reanalysis.

Similar precipitation characteristics as in the CTL simulation are present in the PIN model run (Fig. 2d). The overall bias in the EASM domain is lower than in CTL with $-0.2$ mm d$^{-1}$ but comparable in the YRV ($-1.3$ mm d$^{-1}$, Table 2). With an overall RMSE of 2.9 mm d$^{-1}$ in PIN, the precipitation field is more different to ERAI than CTL (RMSE of 2.4 mm d$^{-1}$). The precipitation maximum along the southern edge of the Tibetan Plateau is more confined in the PIN run compared to CTL but lacks the maximum over Bangladesh ($\geq 12$ mm d$^{-1}$) apparent in ERA-Interim. The PIN simulation shows a precipitation maximum in the BoB similar to ERAI (Fig. 2d). Over the YRV and large parts of southeast Asia and the western Pacific, precipitation in the PIN simulation is underestimated (Fig. 2e, red shading), partly exceeding 3.0 mm d$^{-1}$. In the YRV, the PIN simulation shows values of 3–7 mm d$^{-1}$ (Fig. 2d), with slightly overestimated precipitation in the north and underestimated

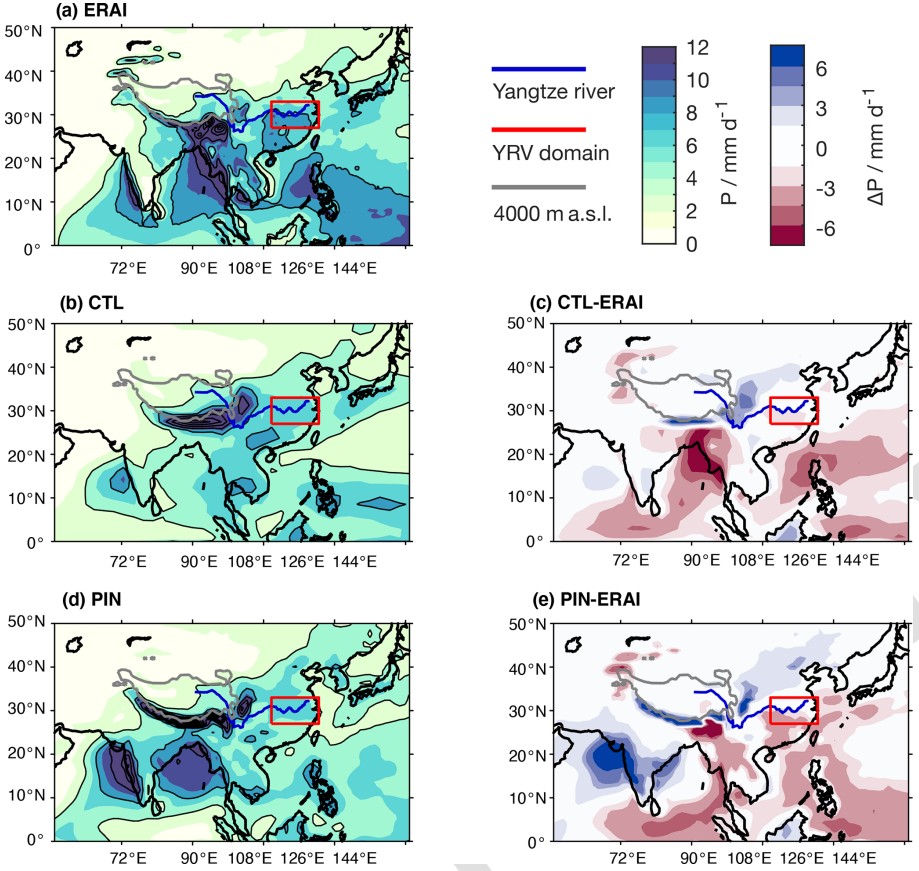

**Figure 2.** East Asian summer monsoon (MJJ) 10-year mean precipitation for **(a)** ERA-Interim (1980–2016), **(b)** CTL (NorESM1-M, 2001–2010), and **(d)** PIN (CAM5.1, 0010–0019) in millimetres per day (mm d$^{-1}$, shading). Solid precipitation contours are shown every 5 mm d$^{-1}$. Precipitation anomalies in comparison to ERA-Interim are shown in panel **(c)** for CTL-ERAI and **(e)** PIN-ERAI (shading, mm d$^{-1}$). The YRV domain is outlined by a red box. The Yangtze River is denoted as a thick blue line, and elevation above 4000 m a.s.l. is indicated by a grey contour.

**Table 2.** Differences of MJJ precipitation and moisture source contribution between model simulations and time slices for the YRV and the entire analysis domain expressed in terms of RMSE and bias (mm d$^{-1}$).

|  | YRV RMSE | YRV bias | EASM domain RMSE | EASM domain bias |
|---|---|---|---|---|
| $P_{CTL-ERAI}$ | 1.536 | −1.391 | 2.395 | −0.989 |
| $P_{PIN-ERAI}$ | 1.861 | −1.340 | 2.917 | −0.220 |
| $P_{LGM-PIN}$ | 0.839 | 0.298 | 2.236 | −0.768 |
| $P_{RCP-CTL}$ | 0.441 | −0.154 | 0.588 | 0.123 |
| $\epsilon_{CTL-ERAI}$ | 0.081 | 0.015 | 0.046 | −0.006 |
| $\epsilon_{PIN-ERAI}$ | 0.150 | −0.128 | 0.051 | −0.021 |
| $\epsilon_{LGM-PIN}$ | 0.034 | 0.030 | 0.020 | 0.001 |
| $\epsilon_{RCP-CTL}$ | 0.049 | −0.027 | 0.019 | −0.004 |

precipitation south of the Yangtze River compared to ERA-Interim.

The seasonal cycle of precipitation in the YRV region has a precipitation mean above 5 mm d$^{-1}$ from April to August and a clear precipitation peak in June according to ERA-Interim (Fig. 3a, dashed black line). Precipitation in the CTL (red) and PIN (blue) simulations peaks during May and June, with a marked decrease from July. Precipitation in May to June is overestimated, while precipitation in July and August is underestimated. Taking into account inter-annual variations (shading), the overall magnitudes and timing of mean precipitation are rather similar in the YRV region for both model runs compared to ERA-Interim, with an RMSE of −1.1 mm d$^{-1}$ for CTL and 1.5 mm d$^{-1}$ for PIN. However, given the coarser resolution, different parameterisations, and absence of data assimilation, a perfect match between the climate models and reanalysis cannot be expected. In addition, only 10-year periods of the CTL and PIN simulations are compared to the longer reanalysis data.

In summary, the representation of summer precipitation in South Asia by the climate model simulations is slightly underestimated, especially in July. The PIN simulation, which

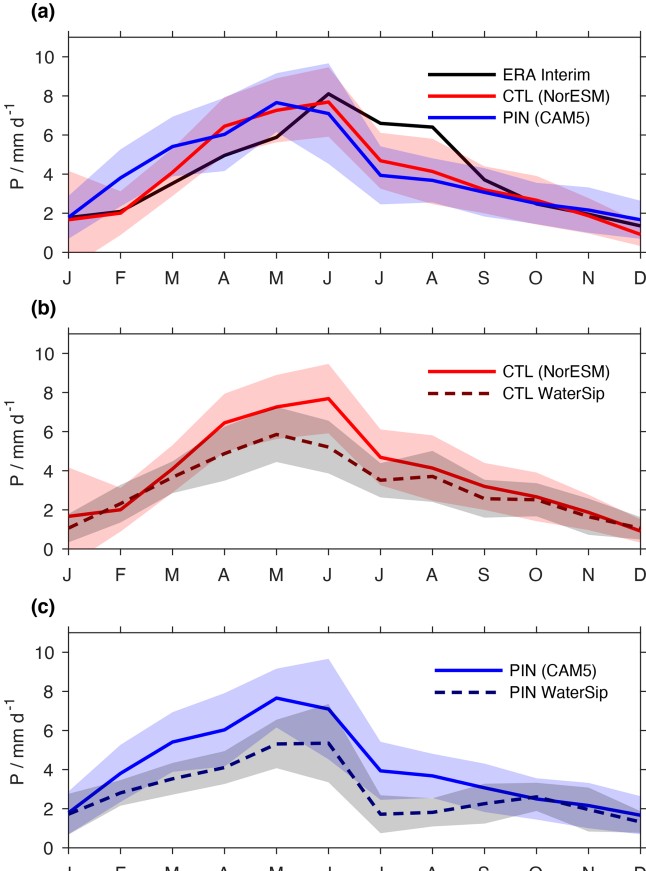

**Figure 3.** YRV monthly mean precipitation in the near-present time period. **(a)** Mean precipitation seasonality for ERA-Interim (black), CTL (red), and PIN (blue) in millimetres per day (mm d$^{-1}$). **(b)** Comparison between simulated CTL precipitation in the YRV (red solid) and the corresponding precipitation estimate from the WaterSip diagnostic (red dashed). **(c)** Comparison between simulated PIN precipitation in the YRV (blue solid) and the corresponding precipitation estimate from the WaterSip diagnostic (blue dashed). Shaded areas indicate the inter-annual 1 standard deviation ($\sigma$) of the mean.

has a higher resolution than CTL, correctly shows a precipitation peak during June. Generally, the precipitation maximum south of the Tibetan Plateau is overestimated, while at the same time precipitation over southern China, the Indochina Peninsula, the BoB, and the western Pacific is underestimated. In particular, the precipitation differences over land should be kept in mind for the later analysis, as they can contribute to the YRV through continental recycling of moisture from land evaporation (Fremme and Sodemann, 2019). Nonetheless, the mean precipitation differences in the YRV are generally smaller than in the surrounding regions, providing a suitable basis for the following analysis of the moisture sources in both climate model simulations.

### 3.2 Precipitation estimate from the WaterSip method for CTL and PIN

The above comparison of climate model precipitation with the ERA-Interim reanalysis shows the skill and shortcomings of both climate models in representing YRV precipitation. However, as described in Sect. 2.3, the moisture sources are obtained from specific humidity fields, rather than using precipitation calculated by the climate models. Furthermore, the WaterSip method provides an estimate of the precipitation amount from the decrease in specific humidity during the last time step before a trajectory end point, denoted here as the Lagrangian precipitation estimate $\Pi$ (in units of mm d$^{-1}$). Differences between this precipitation estimate from Water-Sip and the model-derived precipitation field allow us to assess the representativeness of the results from the moisture source diagnostic. Past studies found that $\Pi$ often has a positive bias of up to 20 %, which could be due to both the neglect of microphysical processes and uncertainty from interpolation during trajectory calculations (Stohl et al., 2005; Sodemann et al., 2008; Sodemann and Zubler, 2010; Sodemann, 2020). Being used as a measure of consistency, the precipitation estimate $\Pi$ should primarily be compared to precipitation from each respective model simulation.

For the CTL simulation, mean estimated precipitation (Fig. 3b, solid red line) is underestimated from April to June compared to simulated precipitation (dashed red line), with an average bias of 0.7 mm d$^{-1}$. All other months show a good correspondence, with an annual average overestimation of about 0.1 mm d$^{-1}$. Estimated precipitation peaks in the same month (May) as the climate model precipitation. For the PIN simulation, estimated precipitation from Water-Sip (Fig. 3c, blue line) is again similar to CAM5.1 precipitation (dashed blue line), with an average bias of 1.2 mm d$^{-1}$. As for the CTL simulation, the summer precipitation is underestimated more than the winter precipitation. On average, CAM5.1 and WaterSip-estimated precipitation differ by 0.5 mm d$^{-1}$, which is within the range seen in previous studies using Lagrangian moisture source diagnostics. Note that also in terms of spatial distribution, the WaterSip-estimated precipitation is also similar to both CTL and PIN precipitation (not shown). The overall good correspondence between the precipitation estimate and climate model precipitation, apart from expected biases, confirms that the choice of parameters for the moisture source diagnostic (Appendix B) allows us to obtain representative insight into the moisture transport and moisture sources of simulated YRV precipitation.

### 3.3 Moisture source locations for near-present simulations

Now we compare the moisture source locations between different climate simulations for the present day and the pre-industrial period. Moisture source area maps can be inter-

preted as the share of total evaporation in the shaded regions that will contribute to precipitation in the target region, the YRV (Fig. 4, red box). These evaporation contributions, or moisture sources, are denoted here by the symbol $\epsilon$ (mm d$^{-1}$). The 35-year mean moisture sources obtained from ERAI during summer (MJJ) serve again as a reference in a comparison between the near-present simulations. Based on ERAI, the moisture sources pertaining to the YRV precipitation are distributed over a fairly large region, reaching across the Indian subcontinent (Fig. 4a). Note that evaporation contributions are clearly lower everywhere ($< 0.9$ mm d$^{-1}$, Fig. 4a) than mean precipitation ($\approx 6$ mm d$^{-1}$), indicating that only a fraction of the precipitation is recycled into the YRV region. For ERAI, the source maximum is just southwest of the YRV. This maximum region contributes with a summer average of 0.8 mm d$^{-1}$ to YRV precipitation. The dotted contour lines denote the 50th and 80th percentiles of $\epsilon$ (Fig. 4c). The innermost dotted contour shows that 50 % of the moisture comes from land regions south and southwest of the YRV and nearby ocean regions. The cyan contour shows that an additional 30 % of moisture comes from more distant land regions as well as parts of the western Pacific, the South China Sea, and the Bay of Bengal. Here, contributions from evaporation to YRV precipitation are generally low ($< 0.2$ mm d$^{-1}$) but spread out over a wide area.

At first glance, the moisture sources for CTL are quite similar to the ERAI results (Fig. 4b). The overall bias compared to ERAI in the EASM domain and the YRV is 0.0 mm d$^{-1}$. Highest values are within or close to the YRV, and the sources extend further south than north and more to the west than to the east, especially over land. Moisture sources for the CTL run show a maximum contribution from a single region of $< 0.9$ mm d$^{-1}$ (Fig. 4b), similar to ERAI. However, some important differences can be seen. For CTL, the 50th percentile extends less south over the Indochina Peninsula than for the reference run, and the 80th percentile extends less to the west and south but more east. For the PIN case, the average moisture contribution from a particular region does not exceed 0.7 mm d$^{-1}$ (Fig. 4c), which is lower than for CTL and ERAI ($< 0.9$ mm d$^{-1}$). The 50th percentile for PIN is similar to the CTL run, encompassing land regions to the south and southwest as well as nearby ocean regions. The 80th percentile in PIN, compared to both CTL and ERAI, is shifted from the western Pacific towards India and into the Arabian Sea compared to CTL, although extending less west and south than ERAI.

Comparing moisture source differences between the CTL and ERAI, it is apparent that the YRV region itself contributes almost the same in CTL as in ERAI (Fig. 4c). The overall bias compared to ERAI in the EASM domain is 0.0 and $-0.1$ mm d$^{-1}$ in the YRV. The largest relative differences are located to the south, outside the YRV region. Using the 80th percentile contour, we focus on differences within the most relevant moisture source regions. For the CTL simu-

lation, evaporation contributions are higher over the South China Sea, the western Pacific, and Bangladesh (Fig. 4c), reaching above 0.2 mm d$^{-1}$ near Hong Kong. There are also regions that contribute less in CTL than in ERAI, in particular over the Indochina Peninsula and to a lesser degree over southern India. For PIN, there is only a small area with increased evaporation contribution over the Bay of Bengal (Fig. 4e). $\epsilon$ is lower by more than 0.2 mm d$^{-1}$ in PIN than in ERAI over the Indochina Peninsula, stretching all the way into the southern YRV. Pacific contributions during PIN are more similar to ERAI than the CTL simulation and thus show only negligible differences.

The larger contribution of eastern sources and smaller land contributions in CTL and PIN could be due to a weaker influence by the Indian monsoon circulation on the YRV in that simulation. Such a circulation difference could explain the smaller contribution from distant western sources and the lower precipitation in summer. However, the larger contribution from the BoB does not fit with this hypothesis. Instead, it is possible that both CTL and PIN are associated with lower rainout over Indochina along the transport pathway, resulting in larger intermediate transport from the BoB. Correspondingly, precipitation recycling could be stronger in ERAI than both climate models, as indicated by Fig. 2, thereby discounting some of the more remote moisture sources. Finally, larger contributions from easterly sources could also be related to circulation differences in terms of a stronger influence by the northwestern Pacific Subtropical High in the CTL simulation.

It should also be noted again here, as mentioned in Sect. 2.1, that the CTL simulation is about 1 K colder than ERAI over the YRV, while PIN is similar to or slightly warmer than ERAI (Fig. 1b). The relatively low temperatures in the YRV persist throughout the entire simulation covering the 2006–2070 period; thus, the comparison between simulations for CTL and RCP (see Sect. 3.6 below) will be internally consistent but potentially influence the moisture sources during both runs.

## 3.4 Moisture source characteristics for near-present simulations

We now compare the seasonal cycle of several geographical, meteorological, and method characteristics at the moisture sources on a monthly timescale for the CTL and PIN simulations, using ERAI as a reference. The first characteristic is the fraction of land area at the moisture sources, weighted by the relative contribution to YRV precipitation (Fig. 5a). On average, the land fraction is close to 70 % for all three simulations. During August and September, the fraction of moisture sources on land is slightly lower for CTL (50 %, red) than for ERAI (55 %, black) and PIN (62 %, blue). The overall bias between CTL and ERAI is $-4.8$ %, compared to $-3.0$ % between PIN and ERAI. The substantially larger contribution from the western Pacific in CTL, as noted above, appears to

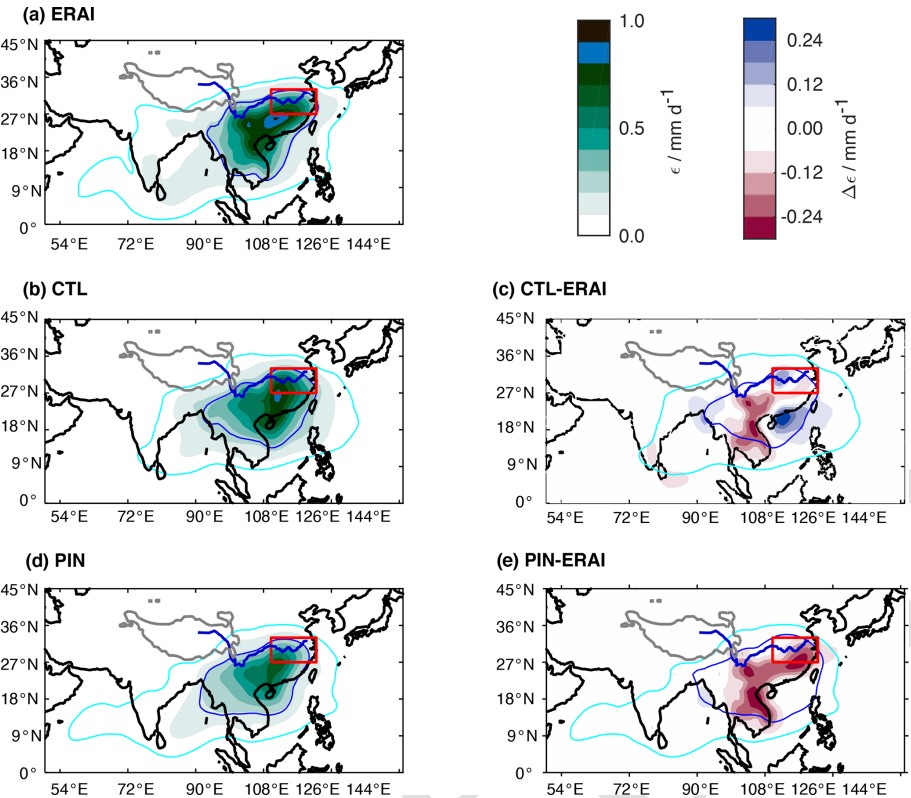

**Figure 4.** Comparison of summer monsoon (MJJ) moisture sources for near-present climate in the YRV. **(a)** Moisture source contributions for ERA-Interim, $\epsilon_{ERAI}$ (shading, mm d$^{-1}$). **(b)** Moisture source contributions for CTL, $\epsilon_{CTL}$ (shading, mm d$^{-1}$); **(d)** moisture source contributions for PIN, $\epsilon_{PIN}$ (shading, mm d$^{-1}$); **(c)** moisture source anomaly for CTL, $\epsilon_{CTL} - \epsilon_{ERAI}$ (%); and **(e)** moisture source anomaly for PIN, $\epsilon_{PIN} - \epsilon_{ERAI}$ (shading, mm d$^{-1}$). Solid contours denote the 50th (blue) and 80th percentile (cyan) of the total water mass. The YRV domain is outlined by a red box. The Yangtze River is denoted as a thick blue line, and elevation above 4000 m a.s.l. is indicated by a grey contour.

have a small influence on the balance between land and ocean sources during July and August (red line).

The mean source longitude and latitude of the moisture sources show a clear albeit weaker seasonality for CTL (Fig. 5c and e). In June and July, the moisture sources are located the furthest south and west in all three runs. While the mean moisture source longitude is similar in all runs (RMSE < 2.5° E), ranging between 95 and 115° E, the mean source latitude shows a bias of 1.2° latitude for PIN and for 2.0° for CTL compared to ERAI (dashed black line), with moisture sources being further south in ERAI. This difference partly translates into differences of the mean moisture source distance (Fig. 5d). Here, ERAI moisture sources are more distant than in CTL (bias -133 km) and PIN (bias −201 km), with the largest differences apparent in February to March and August to September, when the PIN sources (1250 km) are closer than for CTL (1700 km) and ERAI (1750 km), and in June, when the moisture sources are closer in CTL (2000 km) than in PIN (2350 km) and ERAI (2500 km). The WaterSip diagnostic was able to attribute a markedly higher fraction of the estimated precipitation of CTL (bias 2 %) and PIN (bias 1.4 %) to sources than for

ERAI, in particular during May to November (Fig. 5f). The lower fraction accounted for in ERAI can be due to interpolation errors from the trajectory calculation, due to a fraction of moisture that evaporated more than 15 d back in time, and, most importantly, from updates to the humidity field during data assimilation.

Overall, we note a relatively high degree of consistency between the two simulations CTL and PIN with the ERAI dataset. The most substantial differences for CTL are a tendency towards more local sources and a smaller land contribution. PIN has very similar transport characteristics to ERAI overall but a shorter monsoon season with less long-range transport.

## 3.5 Precipitation and moisture sources in the LGM simulation

During the LGM, the global mean temperature was approximately 4 °C colder than pre-industrial temperature (Annan and Hargreaves, 2013). In the CAM5.1 runs, YRV temperature is 4.0 °C colder during the LGM than pre-industrial (Fig. 1b). Given these temperature changes, the hydroclimate

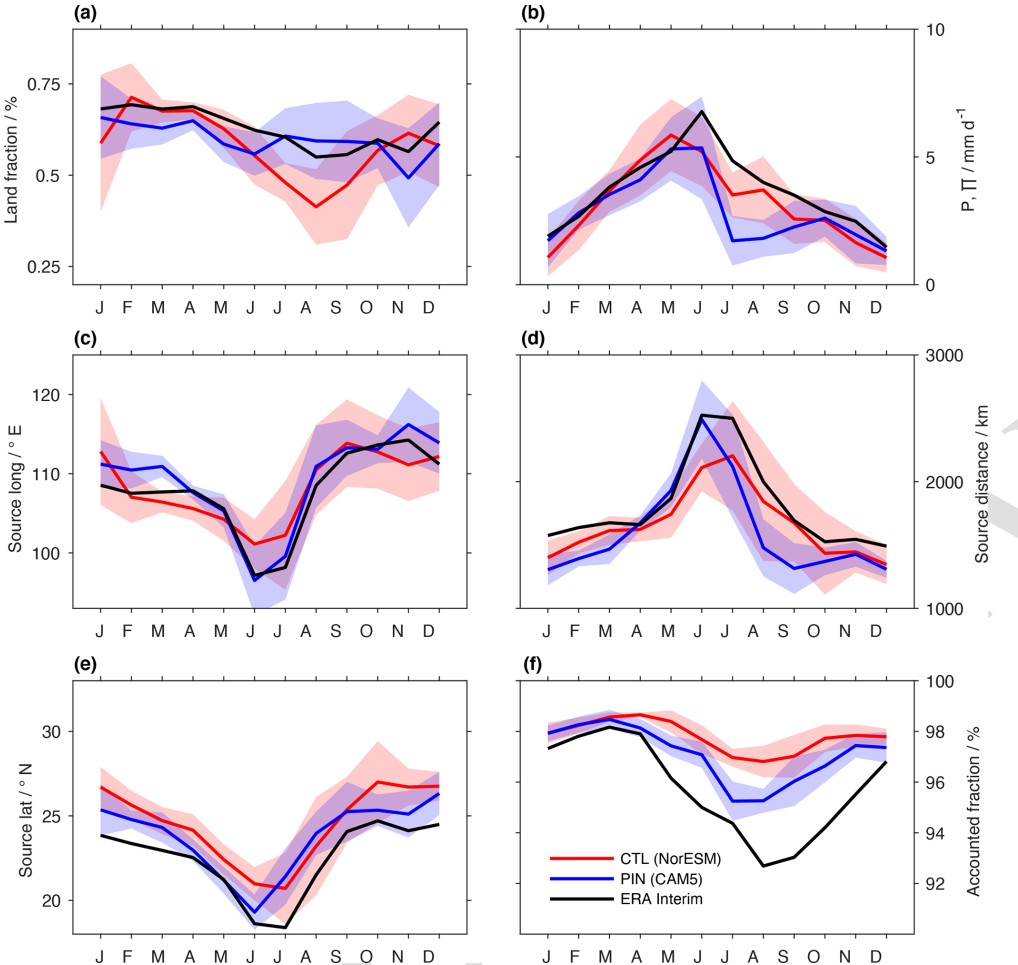

**Figure 5.** Seasonal mean moisture source characteristics for the YRV. **(a)** Land source fraction (%) for CTL (red), PIN (blue), and ERAI (black) with inter-annual 1 standard deviation ($\sigma$) of the mean (shading). **(b)** Estimated precipitation amount (mm d$^{-1}$), **(c)** moisture source longitude (° E), **(d)** moisture source distance (km), **(e)** moisture source latitude (° N), and **(f)** fraction of estimated precipitation accounted for by corresponding moisture sources (%).

of the YRV in terms of atmospheric circulation, moisture transport, and precipitation amounts can also be expected to differ markedly. In previous LGM simulations, the East Asian summer monsoon has been found to be weaker as a result of circulation changes (Jiang and Lang, 2010). Furthermore, the precipitation response to a change in monsoon circulation strength can be expected to vary for different locations and time (Wan et al., 2011). The moisture source perspective adopted here will shed light from a different viewpoint on such expected changes.

Summer precipitation over South and East Asia in the LGM simulation from CAM5.1 is highest along the southern slope of the Himalayas (Fig. 6a). The LGM simulation shows lower precipitation along a belt extending from east to west the Arabian Sea, India, and the BoB compared to PIN (Fig. 6c). At the Indian west coast, LGM precipitation is more than 10 mm d$^{-1}$ lower than in PIN. Closer to the YRV,

over southern China and nearby ocean regions, LGM precipitation is up to 4 mm d$^{-1}$ higher than in PIN.

Summer precipitation values within the YRV region range between 5–7 mm d$^{-1}$ (Fig. 6a), similar to the PIN simulation. While precipitation differences in the YRV are small between LGM and PIN (bias 0.3 mm d$^{-1}$, Table 2), a southward shift of the precipitation maximum can be noted (Fig. 6c). This southward shift is probably related to the overall precipitation increase south of the YRV in the LGM simulation. In the EASM domain there is an overall decrease in precipitation (bias $-0.8$ mm d$^{-1}$). YRV precipitation from the LGM and PIN run shows little change in the seasonal cycle of CAM5.1 precipitation (Fig. 6e). Interestingly, LGM precipitation is higher during July and August than in the PIN run, reflecting a slower onset and overall broader monsoon season. The possible reasons behind this unexpected change in precipitation seasonality and length of the monsoon season are further discussed in Sect. 4.1.

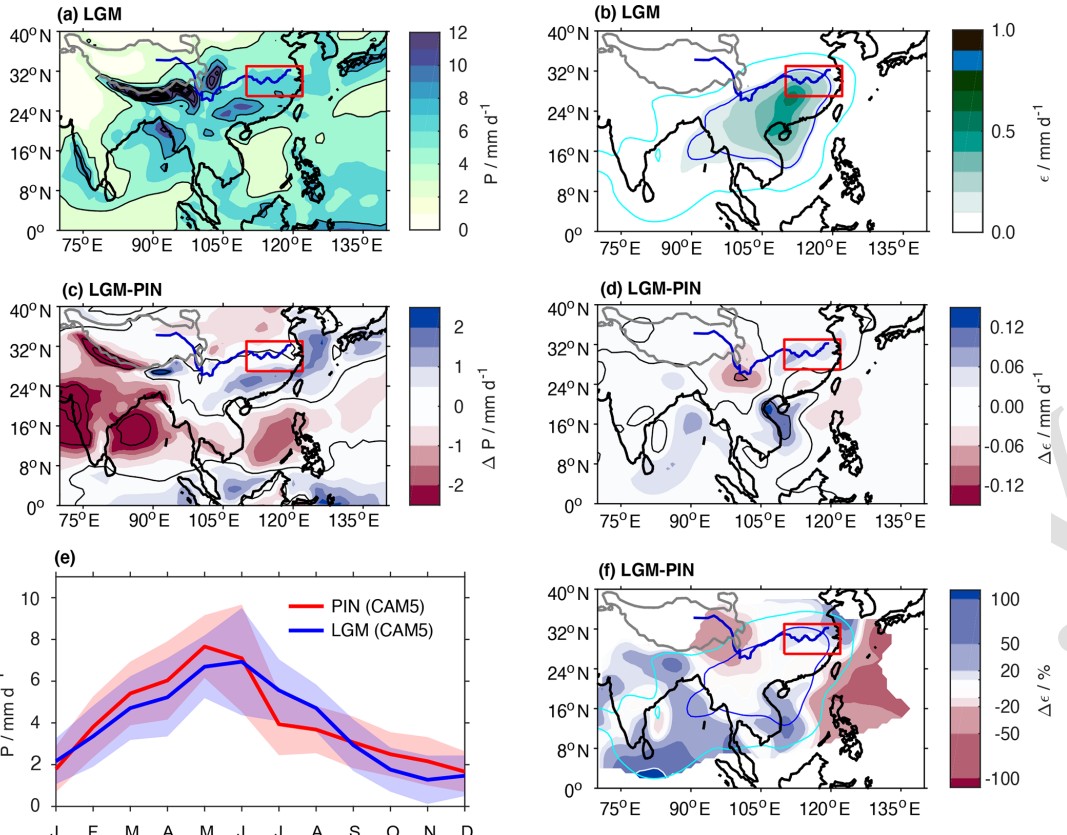

**Figure 6.** Precipitation and moisture source changes for the summer monsoon (MJJ) during the LGM simulation. **(a)** LGM precipitation mean during MJJ (shading, mm d$^{-1}$, and contours every 5 mm d$^{-1}$), **(b)** LGM moisture source contributions ($\epsilon_{LGM}$, shading, mm d$^{-1}$) for MJJ, **(c)** precipitation difference $P_{LGM} - P_{PIN}$ (shading, mm d$^{-1}$, and contours every 5 mm d$^{-1}$) for MJJ, and **(d)** change in moisture source contributions between LGM and PIN ($\Delta\epsilon$, mm d$^{-1}$). **(e)** Mean YRV precipitation seasonality for PIN (red) and LGM (solid). **(f)** Relative change in moisture source contributions between LGM and PIN (%, shading) for areas where $\epsilon_{PIN}$ or $\epsilon_{LGM} > 0.025$ mm d$^{-1}$. Solid contours in panels **(b)** and **(f)** denote the 50th (blue) and 80th percentile (cyan) of the total water mass. The YRV domain is outlined by a red box. The Yangtze River is denoted as a thick blue line, and elevation above 4000 m a.s.l. is indicated by a grey contour.

The moisture sources for the LGM case show the same general features as seen for the near-present simulations (Fig. 6b). The region with highest contributions is located southwest in the YRV region, extending further in a south-
westerly direction. When comparing the LGM sources with moisture sources during PIN, the strongest absolute increase can be seen over the South China Sea (Fig. 6d). Moisture sources over the Bay of Bengal increase, whereas a decrease in $\epsilon$ can be seen over land regions west of the YRV. The bias
in the entire EASM domain and in the YRV is very close to 0.0 mm d$^{-1}$. Comparing LGM and PIN differences as a percentage of moisture sources at each grid point underlines the change to more ocean and less land during the LGM run. Most regions show local increases in moisture contribution
$\epsilon$ to the YRV of 20 % and more (Fig. 6f, blue shading). The strongest increase within the 80th percentile occurs south of India, where contributions to YRV precipitation almost double compared to PIN. In contrast, land regions to the west of

the YRV and a part of the western Pacific show a 20 %–50 % decrease.                                                                                                20
We now compare the seasonal cycle of the YRV moisture source characteristics between the LGM and PIN (Fig. 7). While land fraction is overall similar for the LGM and PIN (bias 0.7 % and RMSE 3.9 %), notable changes include a slightly lower land fraction in the LGM run than in PIN  25
from June to October and an increase in November and December (Fig. 7a). The mean moisture source position is located further south throughout the year in the LGM simulation (bias $-0.7°$ latitude) and during August also further east (bias $-1.2°$ longitude, Fig. 7b and d). These shifts in loca-  30
tion result in more distant moisture sources during August and September (Fig. 7c). On an annual average, moisture sources are therefore slightly more distant in the LGM than PIN (bias 57.7 km). Overall, differences in LGM and PIN moisture source characteristics are remarkably small in rela-  35
tion to the seasonality of each characteristic during the sum-

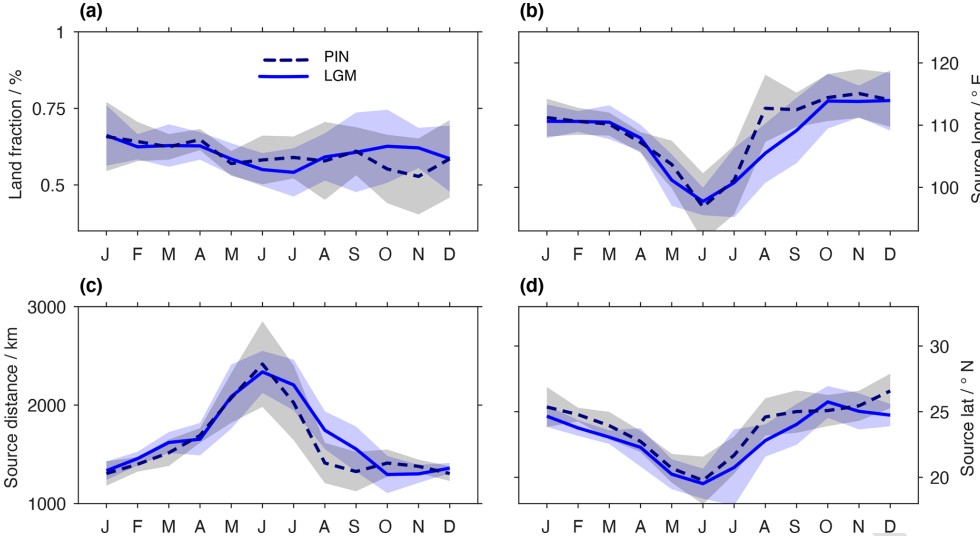

**Figure 7.** As in Fig. 5 but for the moisture source characteristics diagnosed from the simulations LGM and PIN. **(a)** Land source fraction (%) for PIN (dashed black) and LGM (solid blue) with inter-annual 1 standard deviation ($\sigma$) of the mean (shading). **(b)** Moisture source longitude (° E), **(c)** moisture source distance (km), and **(d)** moisture source latitude (° N).

mer monsoon, despite the pronounced temperature changes between both runs.

## 3.6 Precipitation and moisture sources in the RCP simulation

Next, we shift our focus to the simulations of a warmer climate. Multi-model mean CMIP5 results show a wetter East Asian monsoon region towards the end of the century under the RCP6 scenario, with a 7 % average precipitation increase over the whole East Asian monsoon domain and 10 %–15 % locally over the major monsoonal front region (Seo et al., 2013). However, such a change in precipitation is not found consistently between different models (Kitoh, 2017; Yu et al., 2018).

For the future climate simulation RCP analysed here, the general picture of summer precipitation over South and East Asia shows maxima at the southern edge of the Tibetan Plateau and the west coast of India, with second-order maxima over the Indochina Peninsula and the western Pacific, similar to the CTL simulation (Fig. 8a). In absolute terms, summer precipitation decreases in nearby land regions and increases across a wide belt stretching from India to the Philippines (Fig. 8c). For the EASM domain, there are some regional differences that mostly balance out (RMSE 0.6 mm d$^{-1}$, bias 0.1 mm d$^{-1}$). In the YRV there is less precipitation in RCP than CTL (bias $-0.2$ mm d$^{-1}$). Interestingly, it is hardly possible to detect changes in the seasonality of monthly mean precipitation between CTL and RCP, with respect to both amount and timing (Fig. 8e).

Moisture sources for the RCP are centred southwest of and within the YRV, similar to the CTL simulation (Fig. 8b). The largest absolute increase can be seen southeast of the YRV

towards the Taiwan Strait (Fig. 8d). There are small negative biases close to 0.0 mm d$^{-1}$ in both the EASM domain and the YRV, corresponding to smaller moisture source contributions. Comparing the percentage change between CTL and RCP within the 80th percentile contour further highlights the most marked differences, namely an increase of up to 50 % from the east Taiwan Strait towards the ocean regions of the western Pacific (Fig. 8f). The largest decreases are identified northeast of the YRV (20 %–50 %) and over the southern BoB (20 %–50 %).

Comparing the moisture source characteristics of the RCP run to CTL shows that the 10-year climatologies remain mostly within inter-annual standard deviations (Fig. 9). Changes in contribution from moisture sources on land and ocean in the RCP run during summer appear to balance each other, leading to only a slight increase of land contribution in the RCP run in the latter half of the year (Fig. 9a). The most notable change can be seen for the moisture source distance (bias $-71.5$ km, Fig. 9c). During the second half of the year (July to November), average moisture distance decreases from around 1800 to 1500 km, indicating a stronger role of local evaporation and thus of local recycling over land in the RCP run. This is also reflected in more easterly moisture source locations during RCP (bias 0.7° longitude, Fig. 9b) and the slightly more southerly moisture sources in CTL (bias 0.1° latitude, Fig. 9d).

In summary, changes in simulated YRV precipitation are surprisingly small between the RCP and CTL simulations. To first order, this indicates no major reorganisations of the hydroclimate in response to the RCP6 forcings. However, the moisture source characteristics reveal underlying changes in mechanisms, pointing towards a larger contribution of land

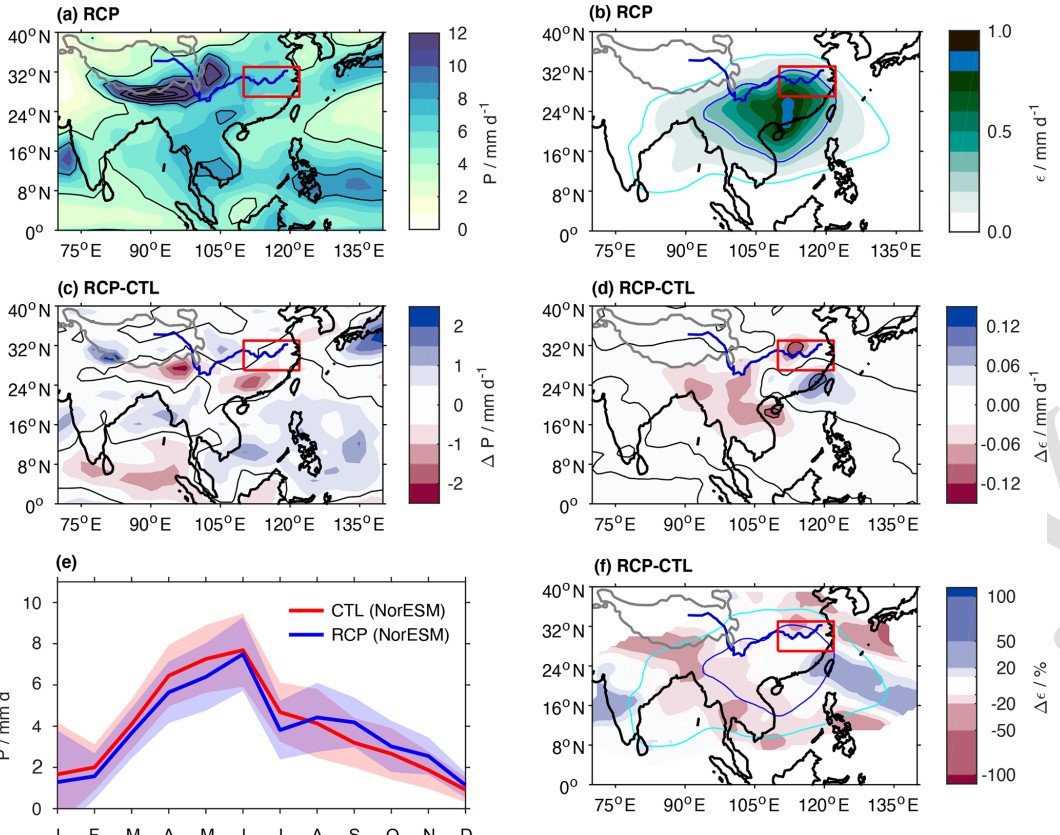

**Figure 8.** Precipitation and moisture source changes for the summer monsoon (MJJ) during the RCP simulation. **(a)** RCP precipitation mean during MJJ (shading, $\mathrm{mm\,d^{-1}}$, and contours every $5\,\mathrm{mm\,d^{-1}}$), **(b)** RCP moisture source contributions ($\epsilon_{\mathrm{RCP}}$, shading, $\mathrm{mm\,d^{-1}}$) for MJJ, **(c)** precipitation difference $P_{\mathrm{RCP}} - P_{\mathrm{CTL}}$ (shading, $\mathrm{mm\,d^{-1}}$, and contours every $5\,\mathrm{mm\,d^{-1}}$) for MJJ, and **(d)** change in moisture source contributions between RCP and CTL ($\Delta\epsilon$, $\mathrm{mm\,d^{-1}}$). **(e)** Mean YRV precipitation seasonality for CTL (red) and RCP (solid). **(f)** Relative change in moisture source contributions between RCP and CTL (%, shading) for areas where $\epsilon_{\mathrm{RCP}}$ or $\epsilon_{\mathrm{CTL}} > 0.025\,\mathrm{mm\,d^{-1}}$. Solid contours in panels **(b)** and **(e)** denote the 50th (blue) and 80th percentile (cyan) of the total water mass. The YRV region is outlined by a red square, and a thick blue contour denotes the Yangtze River.

sources during the second half of the year in RCP. Such a change could be due to circulation changes, possibly connected to stronger continental rainout (and recycling) upstream of the YRV. The moisture source analysis indicates that the small changes in absolute amount may be the result of a shift from more distant to more local land and ocean source regions. Wet soils have previously been shown to be important in inducing more efficient rainout of inflowing air masses in the soil moisture–precipitation feedback (Schär et al., 1999) and could also cause the more local source contributions in RCP detected here. Still, the moisture source regime in the YRV region only shows marginal changes according to our simulation with an RCP6 emission scenario until 2070. A higher-emission scenario, such as RCP8.5, as well as an analysis of full 30-year climate periods and a wider range of models would be needed to corroborate such a shift from remote to more local source contributions (see also discussion in Sect. 4.2).

## 4 Discussion

### 4.1 Differences of the monsoon seasonality during the LGM

During the analysis of the LGM simulation (Sect. 3.5), an increase in the late summer precipitation was observed, leading to an overall broader seasonality of the monsoon (Fig. 6d). Here we explore two hypotheses that could potentially explain the simulated changes in precipitation by means of the corresponding moisture sources (Fig. 7).

A first hypothesis is that the increase in LGM precipitation in the YRV could be caused by a shift in local moisture transport pathways, specifically by increasing moisture transport from regions south of the YRV to the target area. Such a shift in atmospheric circulation can, for example, be caused by downstream effects of the ice-sheet topography changes between PIN and the LGM simulation or other circulation changes in response to different climate forcings. Stronger

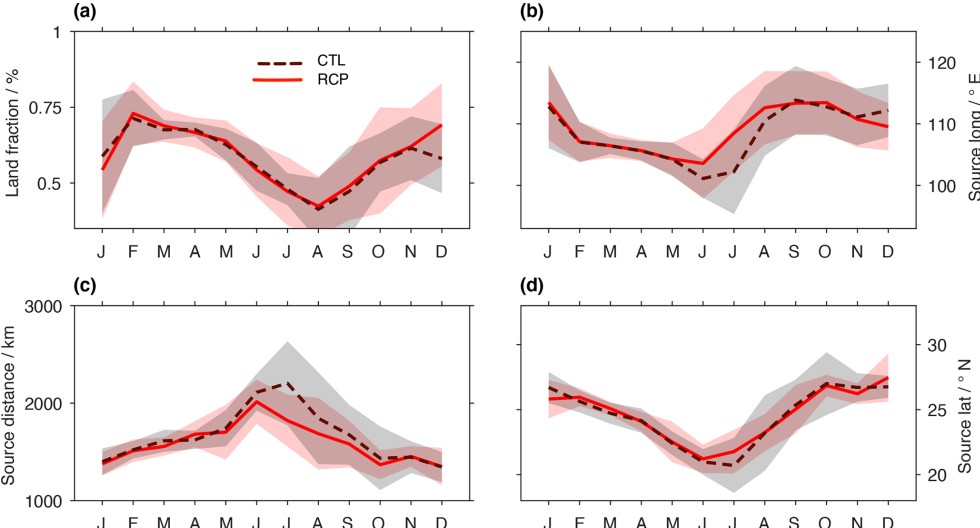

**Figure 9.** As in Fig. 5 but for the moisture source characteristics diagnosed from the simulations CTL and RCP. **(a)** Land source fraction (%) for CTL (dashed black) and RCP (solid red) with inter-annual 1 standard deviation ($\sigma$) of the mean (shading). **(b)** Moisture source longitude (° E), **(c)** moisture source distance (km), and **(d)** moisture source latitude (° N).

southwesterly winds could increase the moisture flux from the South China Sea, the Indochina Peninsula, and the BoB. The higher water vapour flux could be further amplified by more local recycling and thus lead to a general increase of precipitation in the region.

A second hypothesis stems from the apparent opposite relation between the changes in precipitation between the LGM and PIN (Fig. 6b) and the moisture contributed from those regions (Fig. 6d). More specifically, India and the surrounding oceans receive less precipitation but contribute more as sources to YRV. Instead, the western Pacific and western China receive more precipitation but contribute less as moisture sources to the YRV. This finding is most pronounced for the more remote moisture sources. A potential explanation is the soil moisture–precipitation feedback (Schär et al., 1999), where overall colder air masses and land regions during the LGM would be less efficient in raining out the precipitation underway from India and the BoB region and thus lead to increased moisture source contributions from these regions. In other words, the changes in moisture sources could be caused by a decreased efficiency of precipitation processes, leading to a larger contribution from southwestern distant sources.

This second hypothesis is consistent with observations of seasonal variations of moisture transport over central Europe and other regions. For example, Sodemann and Zubler (2010) find that moisture sources are more distant during wintertime than during summer, resulting from both changes in circulation pattern and lower evapotranspiration during the winter. Fremme and Sodemann (2019) highlighted the important role of land regions south of the YRV in the summertime moisture supply during the present-day climate by repeated precipitation recycling. In a colder climate, such in-

direct recycling processes, as well as evapotranspiration, are expected to weaken. Since the first hypothesis would imply stronger evaporation for both recycling and higher water vapour fluxes in the atmosphere, we consider the second hypothesis to be the more plausible explanation for the observed changes from LGM to PIN.

## 4.2 Moisture sources as an indicator for hydroclimatic changes

The comparison between the moisture sources during the near-present climate with LGM (Fig. 6) and the near-present with RCP (Fig. 8) showed mostly scaling of the present-day hydroclimate rather than major reorganisations in response to different climate states. Second-order changes were related to feedbacks with land processes. Here we investigate the implications of this finding more closely. A direct comparison of the 80th percentile contours for all near-present simulations shows that PIN has a moisture source footprint that in terms of extension and shape resembles ERAI quite closely (Fig. 10a). CTL in contrast has a smaller 80th percentile footprint, in particular over the Indian Ocean, India, and the BoB. Also, the 50th percentile contour of CTL extends less to the southwest compared to PIN and ERAI. When comparing, in addition, the LGM and RCP percentile contours, it becomes apparent that the shapes appear to be characteristic for each model (Fig. 10b). In other words, the differences between different models are similarly large as the differences between different boundary conditions.

This implies that NorESM1-M generally simulates the hydroclimate over the YRV differently than CAM5.1, specifically with less long-range transport. Using ERAI as a reference, it appears that NorESM1-M has a bias towards more

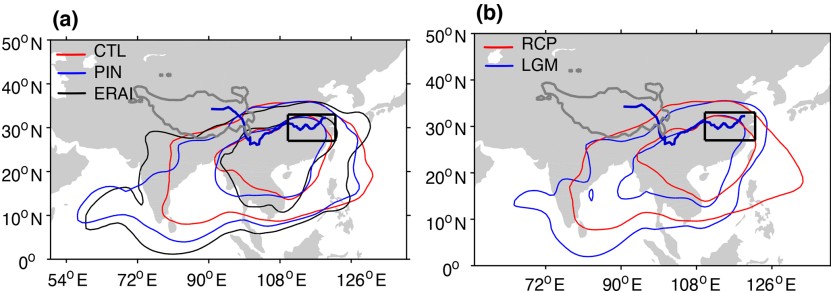

**Figure 10.** Comparison of moisture source changes for **(a)** present-day climates (CTL, dark red; PIN, dark blue; ERAI, black) and **(b)** past and future climate (LGM, blue; RCP, red). Lines show the 50th and 80th percentile of the MJJ mean moisture sources for each respective 10-year simulation. The black box denotes the YRV region; Yangtze River in thick blue. The grey contour denotes an elevation of 4000 m a.s.l.

local moisture sources of the YRV. The reasons for such a bias may have a range of different causes, such as differences in the land-surface model, convection parameterisations, and a coupled versus a slab ocean. Further sensitivity studies re-
5 garding model configuration would be needed to identify the underlying causes for such variability with more certainty. Longer simulation times than feasible for this pilot study are needed to reduce the potential impact of climate variability on these findings. Furthermore, stronger emission scenarios
may be used to detect the full range of sensitivity of the models' hydroclimates.

Finally, the strong resemblance of moisture sources for each model across boundary-condition changes seems to indicate that the models respond primarily by scaling the
15 present-day hydroclimate and to a lesser degree with feedbacks involving the land surface. More substantial reorganisations, such as additional source regions or the shift of the moisture source maximum, are not clearly apparent in our results. This raises the question of to what degree the models
are in fact able to reorganise the hydroclimate and if there are regime shifts to be expected for stronger forcings. If, however, the actual YRV summer monsoon indeed primarily responds to boundary-condition changes by scaling, this knowledge would be immensely valuable to interpret past cli-
mate records and to adapt to future climate change.

## 5   Conclusions

We have investigated the changes in moisture sources and transport processes for precipitation in the Yangtze River Valley (YRV) in climate model simulations across differ-
30 ent climates, including a Last Glacial Maximum (LGM) climate with CAM5.1 and an RCP6 scenario with NorESM1-M, using a quantitative Lagrangian moisture source diagnostic. The changes in the moisture source regions are different from precipitation differences between the model runs and
35 thus provide additional information. Thereby, we gain insight into the how the water cycle in different models responds to boundary-condition changes in the East Asian summer monsoon region, with a focus on the YRV.

Being the first application of this moisture source diagnostic to climate model data rather than reanalysis data, we see
that the same thresholds of the moisture source diagnostic of Sodemann et al. (2008) work for a dataset without increments in humidity from data assimilation. We note an on average larger faction of precipitation that is accounted for in terms of moisture sources, possibly due to better consistency
of model fields in the absence of data assimilation. While for this pilot study time slices of only 10 years have been selected, a more detailed climatology would benefit from using conventional 30-year analysis periods to reduce the impacts of climate variability on the results.

Comparison of a present-day control climate simulation (CTL) with a coupled NorESM1-M and of a pre-industrial climate simulation with an uncoupled CAM5.1 provides a moisture source extent and regional contributions that are overall consistent with the sources found from ERA-Interim
reanalyses. Land contributions from areas south of the YRV and over Indochina are the most important moisture source, whereas ocean areas over the Bay of Bengal (BoB), the South China Sea, and the adjacent Pacific are responsible for most of the remaining contributions. The climate model runs
thereby show slightly more local moisture origins than ERA-Interim.

For the CAM5.1 LGM simulation, moisture sources show small differences to near-present simulation of the same climate model during summer. These differences could be
caused by an increased efficiency in the moisture transport from southwestern distant sources caused by less rainout en route. A more detailed investigation of the respective contribution of forcing changes to the atmospheric circulation would be needed to corroborate these indicative findings.

For the coupled NorESM1-M simulation using the future climate scenario RCP6, the moisture sources show only small differences to the control run. Differences in the moisture source characteristics do not exceed the inter-annual standard deviation of the time slices analysed. The small
change between the moisture sources of the control and fu-

ture climate could be connected to the in general limited change seen in precipitation over the East Asian monsoon region.

The first-order response in terms of moisture sources and moisture transport to the YRV across different climate forcings, from LGM to RCP runs, is a scaling of the hydroclimate, rather than a major reorganisation for the different 10-year time slices analysed here. Moisture sources thereby remain similar to first order with respect to the location, the magnitudes, and the relative contributions of moisture from land and ocean areas. A second-order effect notable from the LGM through to the RCP simulation is a larger emphasis of local land sources in warmer climates. Although relatively small, the more local origins could indicate more efficient rainout of transported water vapour in a warmer climate, with subsequent precipitation recycling. Additional sensitivity studies would be needed to quantify the relative contributions of, among others, grid resolution, the land model, ocean models, and convection parameterisations to the location of moisture source regions. In addition, simulations with a higher-emission scenario could provide stronger forcing to potentially emphasise responses of the hydroclimate. We expect that the moisture source perspective introduced here to climate model data will also provide valuable insight during such sensitivity studies.

Interestingly, some moisture source characteristics show similar or even larger differences in the different climate models than between different simulations using the same model. If we assume the models adequately represent relevant aspects of the Earth system, our study suggests that the hydroclimate responds with scaling rather than major reorganisation to moderate climatic changes over the YRV, while changes in land–atmosphere interaction play a detectable, but secondary, role. Alternatively, we can pose the question of how realistic the representations of climate and of response to boundary-condition changes by the different models are. It is possible that model responses underestimate the potential of larger hydroclimatic reorganisations, for example due to limited feedbacks involving the land surface. It would be valuable to decipher if there are potentially stabilising factors in the East Asian summer monsoon system or if models potentially lack feedbacks that enable more substantial changes in the hydroclimate. Comparison and interpretation of moisture source information with paleoclimate records of stable isotopes could enable us to tie simulated hydroclimatic model responses to observational evidence.

## Appendix A: Changes in CAM5.1 between LGM and PIN

The changes between LGM and PIN during the 10-year time slice considered here for surface air temperature in the EASM region are between $-2$ and $-4\,\mathrm{K}$ over most of the region (Fig. A1a). Some ocean areas over the Indian Ocean and the West Pacific show a temperature decrease of less than $2\,\mathrm{K}$. These changes are similar to those in PMIP2 and CMIP5 (Harrison et al., 2014). Annual mean anomalies of zonal wind speed at 850 hPa show an increase in zonal wind speed of around $1\,\mathrm{m\,s^{-1}}$ across a broad band from northern India to the Philippines in the LGM simulation (Fig. A1b).

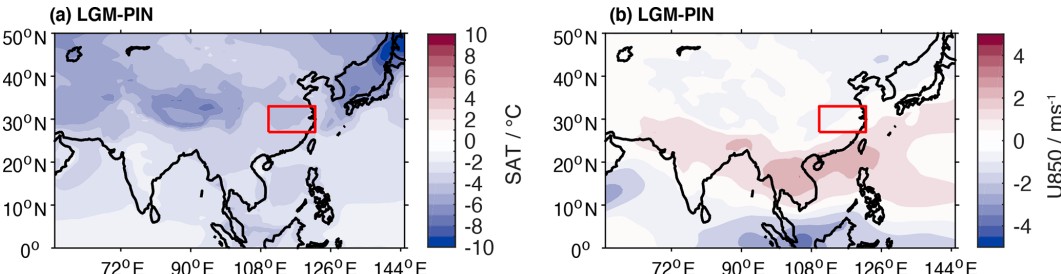

**Figure A1.** Annual mean differences between LGM and PIN simulations in **(a)** surface air temperature (shading, K) and **(b)** zonal wind speed at 850 hPa (shading, m s$^{-1}$) as simulated by CAM5.1. Differences are based on the 10-year time slices from simulation years 0010–0019.

## Appendix B:  Sensitivity to WaterSip thresholds and time-step length

An important advantage when diagnosing moisture sources from the climate model simulations over the ERA-Interim reanalysis is that they are not affected by data assimilation. The absence of humidity increments from data assimilation allows us to analyse the effect of the choice of certain parameters of the WaterSip diagnostic. In this section, the effect of a change of thresholds and time-step length on the moisture sources is exemplified for backward trajectories obtained for 1 year (2002) from the CTL simulation (NorESM1-M). We consider this to be a representative example for trajectories from either set of climate model trajectories. The thresholds tested are the threshold value for the minimum change in specific humidity per time step $\Delta q_c$, the time step $\Delta t$, and the relative humidity threshold RH$_c$. From these threshold tests, we are able to evaluate the robustness of the overall findings presented above.

Throughout this study, the thresholds for $\Delta q_c$ and $\Delta t$ have been set to 0.1 g kg$^{-1}$ 6 h$^{-1}$ and 6 h, respectively. These parameters are the same as for the study of Fremme and Sodemann (2019) using ERAI data, thus making the results directly comparable. The specific humidity change threshold $\Delta q_c$ determines the limit above which changes in specific humidity in air parcels are recorded as due to either precipitation or evaporation events. Humidity changes below the threshold are considered due to interpolation errors and are not taken into account for the moisture source diagnostic. The time steps tested here are for 3, 6, 9, and 12 h duration and for the specific humidity thresholds shown in Table B1.

### B1   Sensitivity to specific humidity threshold changes

The effect of the specific humidity threshold change together with changes of the time-step length on moisture sources becomes apparent from the mean distance of the moisture sources for YRV precipitation (Fig. B1). For comparison, the $\Delta q_c$ values are converted to the unit g kg$^{-1}$ 1 h$^{-1}$. With the setting chosen here for 6 h as a reference (red symbols), it is apparent that longer $\Delta t$ results in more distant sources for the same $\Delta q_c$. The moisture source distance is approximately

**Table B1.** Range of parameters for the sensitivity tests for NorESM1-M for the year 2002, including the time steps $\Delta t$, specific humidity thresholds $\Delta q_c$, and the relative humidity threshold RH$_c$. Numbers in brackets for $\Delta q_c$ are in units of g kg$^{-1}$ h$^{-1}$ to enable an easier comparison.

| Time step length | Specific humidity thresholds (g kg$^{-1}$) | | | |
|---|---|---|---|---|
| 3 h | 0.05 (0.0167) | 0.5 (0.167) | 1.0 (0.333) | |
| 6 h | 0.1 (0.0167) | 0.5 (0.083) | 1.0 (0.167) | 2.0 (0.667) |
| 9 h | 0.1 (0.0111) | 1.0 (0.111) | 2.0 (0.222) | 3.0 (0.333) |
| 12 h | 0.1 (0.008) | 1.0 (0.083) | 2.0 (0.167) | 3.0 (0.250) |
| Relative humidity thresholds (%) | 0   30   50   60   70   80   90 | | | |

300 km larger for a 6 h than for a 3 h time step. Similarly, the sensitivity runs with a 12 h time step also have about 300 km more distant source regions than the 6 h run. Long $\Delta t$ can invalidate the assumption that all humidity changes in an air parcel are due to either evaporation or precipitation only and that the other process can be disregarded. While this would suggest that the 3 h time step gives the most accurate results, such short time steps can introduce larger errors as $\Delta q_c$ reaches a similar magnitude as numerical noise and interpolation errors. Values of above 6 h are likely to lead to degraded trajectory calculation and to overlooking the influence of diurnal variation. From the above discussion, it appears that the moisture source distance has an error margin of several hundred kilometres, or on the order of 10 %– 20 %, depending on the exact parameter choice for the time step. For the same time step, changes in $\Delta q_c$ give a less distant source for higher thresholds. However, threshold values above 0.1 g kg$^{-1}$ 1 h$^{-1}$ are hardly typical and only include very strong evaporation and precipitation events. For $\Delta q_c$ below 0.1 g kg$^{-1}$ 1 h$^{-1}$, the source distance is much less sensitive to the threshold value than to the time step.

### B2   Sensitivity to relative humidity threshold changes

The relative humidity threshold RH$_c$ is another influential parameter of the moisture source diagnostics. RH$_c$ sets the minimum RH at which a moisture decrease within the target

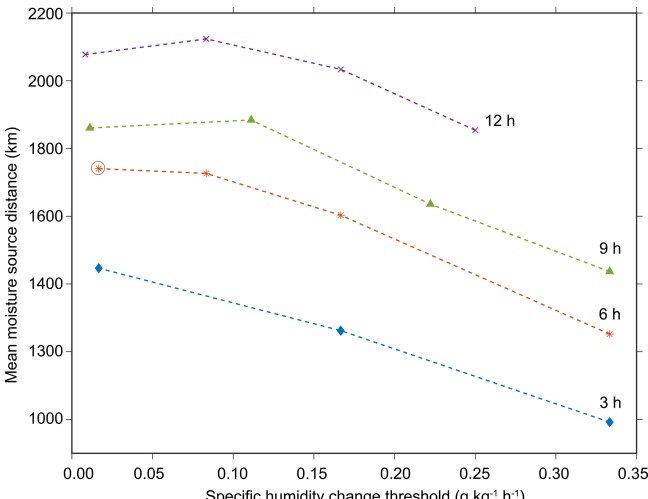

**Figure B1.** Sensitivity of the moisture source distance with respect to changes in the specific humidity threshold ($\Delta q_c$) and time step ($\Delta t$) for a range of 3 to 12 h. Markers denote the average distance from moisture source to the target region in kilometres. All sensitivity runs were done for the year 2002 of the CTL simulation.

region is considered to be a precipitation event. Therefore, $RH_c$ has a large effect on the precipitation estimated by the WaterSip method. However, $RH_c$ can also affect the moisture sources, as the weight of individual uptakes corresponds to estimated precipitation in the target region. Throughout this study, we use a threshold value of $RH_c = 80\%$. In the sensitivity tests discussed in this section, the effect of varying the $RH_c$ between 0%–100% is tested. Note that we thereby only consider the results using $RH_c$ between 70%–90% to be physically consistent with the model parameterisations.

We evaluate the effect of the $RH_c$ changes with respect to the impact on the precipitation estimate for the target region and the moisture source distance. As expected, a more restrictive, higher $RH_c$ leads to a lower precipitation estimate for the target region (Fig. B2a). A change of $RH_c$ from 80% to 70% and to 90% leads to a change in the all-year precipitation mean (3.2 mm d$^{-1}$) by +32% and −43%, respectively. The value of $RH_c = 80\%$ used throughout this study gives the precipitation estimate closest to the all-year mean precipitation from NorESM.

Changes in precipitation with changing $RH_c$ are accompanied by a change in source distance (Fig. B2b). When $RH_c$ is changed from 80% to either 70% or 90%, the mean source distance changes from an average of 1740 km by only −5% and 4%, respectively. Even with a more extreme change in the $RH_c$ to as much as 30%, source distance only changes by around 20% compared to $RH_c = 90\%$.

The small changes in source distance suggest that the $RH_c$ only has a minor impact on the moisture source results presented here and lead to a uniform scaling of the moisture sources rather than a shift. Therefore, we do not find a need to

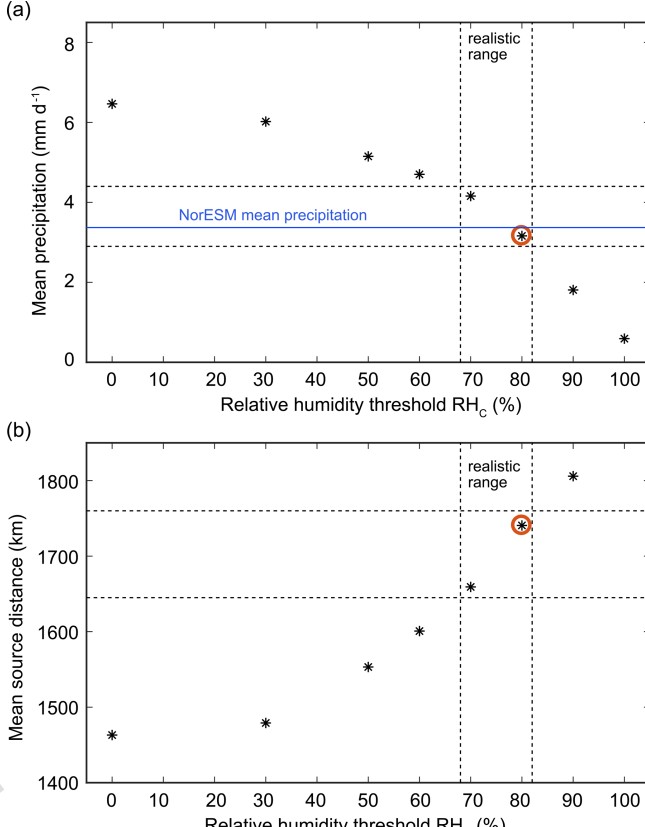

**Figure B2.** Sensitivity of the precipitation estimate and moisture source distance to changes in the relative humidity threshold ($RH_c$). **(a)** Sensitivity of the precipitation estimate (mm d$^{-1}$) in response to $RH_c$ (%). Anticipated realistic range of precipitation indicated by dashed lines. Blue line shows precipitation simulated by NorESM1-M. **(b)** Response of moisture source distance (km) to changes in $RH_c$ (%). Value for threshold $RH_c$ adopted in this study is highlighted by a red circle. All sensitivity runs are based on the year 2002 of the CTL simulation.

modify the $RH_c$ of 80%, in particular, as the precipitation estimated by WaterSip follows the original climate model precipitation reasonably well (Fig. 3). We conclude from this sensitivity study that the moisture source results are overall robust regarding specific parameter changes within physically reasonable limits. This finding confirms the sensitivity experiments of the same method by Läderach and Sodemann (2016) based on ERAI data.

*Code availability.* The FLEXPART model code is available from the flexpart code repository http://www.flexpart.eu/downloads TS4. The WaterSip software code is in preparation for publication in a paper for *Geoscientific Model Development* CE2 and is not yet publicly available.

*Data availability.* The SST and sea ice climatology dataset used for simulating LGM conditions is accessible at https://www.earthsystemgrid.org/dataset/ucar.cgd.ccsm4.b40. lgm21ka.1deg.003M.html TS5. The output data files with moisture source information for different climates are available on Zenodo at https://doi.org/10.5281/zenodo.7907221 TS6.

*Author contributions.* AF and HS developed the concept of the study, led the investigation and formal analysis, and wrote the original draft of the paper. PJH and ØS contributed to the investigation and formal analysis. All authors contributed to the revision and final editing of the manuscript.

*Competing interests.* The contact author has declared that none of the authors has any competing interests.

*Acknowledgements.* This work was supported by the project Norges Forskningsråd grant UTF-2016-long-term/10030 and the Swiss National Science Foundation through grant no. 200021_143436 "Spatial and Temporal Scales of Linkages in the Atmospheric Water Cycle (Waterscales)". This work used storage capacity from the Norwegian computing infrastructure Sigma2 through project NS9054K (COPEWET), NN9555K, and NS2345K (EVA). Access to the ECMWF ERA-Interim reanalysis data was provided through Met Norway. This work has been supported by the Research Council of Norway through the project EVA (grant no. 229771).

*Financial support.* This research has been supported by the Schweizerischer Nationalfonds zur Förderung der Wissenschaftlichen Forschung (grant no. 200021_143436) and the Norges Forskningsråd (grant nos. 229771, NS9054K, NN9555K, and NS2345K). TS7

*Review statement.* This paper was edited by Martin Singh and reviewed by Bo Liu and one anonymous referee.

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

TS7    Please note that there still is a discrepancy ("Norges Forskningsråd grant UTF-2016-long-term/10030"). Please check.

TS8    Please confirm volume number and page range. The information you provided was for a different paper.

TS9    Please confirm volume number and page range as provided on the website.