# Peer review of "Model-simulated hydroclimate in the East Asian Summer Monsoon region during past and future climate: a pilot study with a moisture source perspective"

_Weather and Climate Dynamics, 2022_

## Referee Comment (RC1)

**Review of WCD-2022-52 by Astrid Fremme et al. 2022**

Bo Liu[a]

[a] Department of Atmospheric Science, China University of Geosciences, Wuhan

Email: boliu@cug.edu.cn

**General Comments**

This study investigates the hydroclimate variability of the Yangtze River Valley under different climate states, i.e., present, last glacial maximum, and future projection scenario, from the perspective of moisture sources and transport. The comparison of moisture sources and the associated contributions between modern climate and paleoclimate (LGM) could advance the understanding of the variability in the East Asian Summer Monsoon. Also, as the authors pointed out, it is the first time this moisture diagnostic driven by free-running model results. While I find these results presented very interesting, I have some suggestions for the authors to consider for further improvement.

**Specific Comments**

1. How about the simulation of LGM with CAM5.1? A comparison between this simulation and the LGM experiments, in CMIP5/6, in simulating precipitation and the EASM circulation is necessary.

2. The statements in the abstract ["differences in moisture source conditions are larger between the different climate models than between different climatic boundary conditions using the same model"] made the topic less important. Of course, the evaluation of results driven by climate model output is important. However, when I saw the title, I expected more enlightenment when comparing between past and future climate conditions. I would suggest the authors consider the linkage between the intensity of EASM and the moisture contributions from different areas.

3. Moisture recycling during the transport pathway may influence the contributions of moisture sources. In Fig. 4, moisture contributions from the Bay of Bengal increase in both CTL and PIN simulations, compared with ERAI. I guess such increases are connected to the lack of precipitation (moisture loss) over the Bay of Bengal. In

other words, anomaly moisture gains (lack of moisture losses) over this region in both CTL and PIN simulations increase the corresponding moisture contributions. Moreover, the weaker precipitation in CTL makes the moisture contribution from the Bay of Bengal larger than that in the PIN. The moisture contributions from the Pacific are likewise.

4. L141: The PIN simulations are substantially colder than ERA-Interim?

5. The CTL time slice used in this study is a 10-year period (L168) or a 5-year period (Table. 1). If the 1996-2005 period is selected, 1997/1998 strong El Niño is selected.

6. PIN is pre-industrial simulation and CTL is present simulation. Therefore, the comparison between ERAI and CTL is different from that between ERAI and PIN. Therefore, I would suggest the authors reorganize this manuscript solely with results from LGM and PIN simulations. The comparison between CTL and RCP could be formed into another manuscript.

**Technical Comments**

I would suggest the authors thoroughly check these technical details in this manuscript before submitting the revised manuscript.

1. It is "Yangtze River", not "Yangtse River". There are several "Yangtse River" in the text.

2. The usage of "screen-level temperature" and "near-surface temperature". I would suggest using one of "near-surface temperature" or "surface air temperature".

3. RCP is the abbreviation for "Representative Concentration Pathway", not for "Reference Climate Projection" (caption of Fig. 1).

4. I would suggest the authors only use "Celsius (℃)" to describe the temperature and the associated difference.

5. Some expressions should be uniformly used in this study.

    5.1 "Western Pacific" instead of "West Pacific".

    5.2 "South China Sea" instead of "south China Sea".

    5.3 "Tibetan Plateau" instead of "Tibetan plateau".

6. L288: While the authors say "However, some differences can be seen." The following sentence gives a "similar to ERAI".

7. Some descriptions are with wrong figure numbers.

    7.1 L297: "Fig. 4d" should be "Fig. 4c". Also, "both climate models" should be CTL simulation.

    7.2 L300: "Fig. 4d" should be "Fig. 4c".

    7.3 L325: "Fig. 5b and c" should be "Figs. 5c and 5e"?

    7.4 L372: "Fig. 7b and c" should be "Figs. 7b and 7d"?

    7.5 L373: "Fig. 7e" should be "Fig. 7c"?

    7.6 L384: "Fig. 8b" should be "Fig. 8c"?

    7.7 L388: It seems that Fig. 8d does not indicate "and directly to the east towards Bangladesh".

    7.8 L390: "…, coastal regions of the South China Sea and the Bay of Bengal towards India" seems with no significant increase.

8. Figure 7: "RCP" should be "LGM".

---

## Referee Comment (RC2)

**Review of WCD-2022-52 by Astrid Fremme et al. 2022**

This manuscript aims to examine how the hydroclimate of the Yangtze River Valley changes along different climates, from the last glacial maximum to the present, and then under a future projection scenario. This fact is based on the comparison of the moisture sources for the target region. The authors used two model outputs from climate models CAM5.1 and NorESM1-M and ran the FLEXPART model trying to obtain the moisture sources for the Yangtze River Valley. ERA-I was also used.

While the exercise of using the different datasets to run de FLEXPART model is novel, there are several major concerns about the modeling, the data used, calculation, comparison, and interpretation that lead to many doubts to this reviewer; many of them related to the robustness of the statistical analysis of this work.

I believe substantial modifications are necessary and thus I would reject this manuscript if no major changes are made.

My main comments and reasons for this decision are:

1. My biggest concern is the robustness of the statistics. Throughout the paper, there is no statistical analysis for the comparison of the simulations carried out for the present time (control period). It is needed an annual and seasonal comparison between the dataset used and derived variables from the FLEXPART outputs for the control periods. A visual comparison of the obtained fields, which are shown in the manuscript as maps, is not sufficient to conclude whether the moisture sources are similar or not. A statistical study is necessary. Typical statistics are used in this type of analysis: mean absolute error (MAE), root mean square error (RMSE), Pearson's correlation (R), Bias (B), standard deviation (STD), and coefficient of determination and variation.

2. My other big comment is about the selection of the length of the period to compare. What is the reason for the selection of these 10 years (in the table appears that is 5 years…)? Does decadal variability play some role in the results obtained? The author comment that the period is neutral, but in the selected period as control exits a strong ENSO event, 1997-98, and during the decade the sign of the Pacific Oscillation (POD) was the same, and it is known that the East Asia Monsoon precipitation is affected by ENSO, but the POD strengths this relationship when positive. The Indian Ocean Dipole, which is the role of this mode over the region? If the hydroclimate variability and changes along the time (past-present-future) of a region where the EASM affects is the goal of this paper, it needs to be into account as the other modes commented. The authors should use 30 years as usual for climatological studies.

3. To resolve these issues, it is necessary to extend the study period as much as possible, compare a longer period, and do statistical analysis.

Overall, a more critical discussion of the limitations and uncertainties of the study seems necessary.

Without these questions (methodological and analytical issues) resolved, this reviewer cannot consider this manuscript for publication at this stage.

Other aspects need more attention by the authors:

4. The input data for running FLEXPART model have different characteristics in terms of vertical and horizontal resolution. Is the number of particles modeled the same in the different experiments? If they are the same, the model preconditions are different, and this fact could affect the results and the interpretation of the field comparison. This needs to be checked and explained in the manuscript.
5. The authors comment that the thresholds in WaterSip were changed, and the selected RH% , for instance, was determined for NorESM dataset. Why not other data or reanalysis? Could this affect the outputs of the model when used for the future? The diagnostic for the imposed thresholds was done only for one year! Which year? Why this one? If the year selected was another, could the threshold change? … How this affect the results?
6. Only one emission scenario, RCP6.0, was used in the study. Why? RCP6.0 represents an intermediate scenario, and it is not very used in climate change studies. What about RCP8.5?
7. Many of the comparisons in the manuscript, such as those derived from figure 3, etc…, were in absolute values. 1.0 mm/day seems a small quantity, but it is not small if the variables to compare have a range from 4 to 8mm/day. Better in %.
8. There are many typo errors in the manuscript (table, figures, captions, …). Please check the manuscript carefully when the revision is done.

This review has further comments on the results and their interpretation, but these are not necessary at this stage of the review until the above is resolved.

---

## Author Response (AR1)

Dear Editor,

we have now completed the revisions in response to the two reviewer comments. We feel we could address all reviewer comments adequately, and made substantial changes throughout the manuscript. In response to the reviewer's comments, both the our main message, reliability of the results, and the overal clarity of the presentation have improved. In the responses below, we add our replies in blue text, starting with the keyword "Reply". We also made a change to Figures 2 and 4 in the manuscript, where we now use absolute units in the comparison panels of the precipitation and moisture sources, since this is more consistent with the statistics in Table 1.

On behalf of all authors,
Harald Sodemann

**Review of WCD-2022-52 by Astrid Fremme et al. 2022**

Bo Liu, Department of Atmospheric Science, China University of Geosciences, Wuhan (boliu@cug.edu.cn)

**General Comments**

This study investigates the hydroclimate variability of the Yangtze River Valley under different climate states, i.e., present, last glacial maximum, and future projection scenario, from the perspective of moisture sources and transport. The comparison of moisture sources and the associated contributions between modern climate and paleoclimate (LGM) could advance the understanding of the variability in the East Asian Summer Monsoon. Also, as the authors pointed out, it is the first time this moisture diagnostic driven by free-running model results. While I find these results presented very interesting, I have some suggestions for the authors to consider for further improvement.

**Reply:** We thank the reviewer for the detailed, thoughtful and constructive comments.Below we state our response to each of the raised points.

**Specific Comments**

1. How about the simulation of LGM with CAM5.1? A comparison between this simulation and the LGM experiments, in CMIP5/6, in simulating precipitation and the EASM circulation is necessary.

**Reply:** We agree that a comparison between the CAM5.1 LGM simulation and CMIP5/6 simulations can provide important context regarding main climatic features, such as precipitation and circulation. In the revised manuscript, we added a new figure to the appendix, and in the text we briefly compare the models climate characteristics seen there in the context of the PMIP2 experiments (Harrison et al., 2014).

2. The statements in the abstract ["differences in moisture source conditions are larger between the different climate models than between different climatic boundary conditions using the same model"] made the topic less important. Of course, the evaluation of results driven by climate model output is important. However, when I saw the title, I expected more enlightenment when comparing between past and future climate conditions. I would suggest the authors consider the linkage between the intensity of EASM and the moisture contributions from different areas.

**Reply:** We see the potential discrepancy between the title and the sentence in the abstract highlighted by the reviewer. In the revision, we have moderated the statement now as "differences in some moisture source characteristics are larger between the different climate models than between different climatic boundary conditions using the same model" and instead highlighted the scaling of the hydroclimate as compared to a reorganisation, and the second-order effects of land-atmosphere interactions. To hopefully better direct reader expectations, we have rephrased the title now as "Model-simulated hydroclimate of the East Asian Summer Monsoon during past and future climates: insights from a moisture source perspective."

We also want to stress that it is a key finding from our study that the hydroclimate appears to be simulated differently between models, and the moisture source diagnostic provides access to this information. We have revised the abstact to better highlight our findings with regard to the role of moisture recycling over land. Nonetheless, we consider our finding that moisture source and transport differences between models are large, and to some extent consistently so across different climates, points to the large uncertainties in model-simulated hydroclimate. Correctly simulating a coupled cycle, such as the water cycle, is as important as it is difficult, and more work is needed to find out which representations are most realistic. We hope that with the changes in the revised manuscript, we have improved the coherence between title and abstract, such that we do not disappoint reader expectations.

3. Moisture recycling during the transport pathway may influence the contributions of moisture sources. In Fig. 4, moisture contributions from the Bay of Bengal increase in both CTL and PIN simulations, compared with ERAI. I guess such increases are connected to the lack of precipitation (moisture loss) over the Bay of Bengal. In other words, anomaly moisture gains (lack of moisture losses) over this region in both CTL and PIN simulations increase the corresponding moisture contributions. Moreover, the weaker precipitation in CTL makes the moisture contribution from the Bay of Bengal larger than that in the PIN. The moisture contributions from the Pacific are likewise.

**Reply:** We agree that the Bay of Bengal region shows consistent differences between the climate models and ERAI. In general, stronger recycling over Indochina, for example, would decrease more remote contributions, thus indicating stronger land recycling in the ERAI simulation than CTL and PIN. In the revised manuscript, we added a brief discussion of the moisture recycling along the transport pathways to the description of Fig. 4:

"The larger contribution of eastern sources in CTL could be due to a weaker influence by the Indian monsoon circulation on the YRV in that simulation. Such a circulation difference could explain the smaller contribution from distant western sources and the lower precipitation in summer. However, the larger contribution from the BoB does not fit to this hypothesis. Instead, it is possible that both CTL and PIN are associated with lower rainout over Indochina along the transport pathway, resulting in larger intermediate transport from the BoB. Correspondingly, precipitation recycling could be stronger in ERAI than both climate models, as indicated by Fig. 2, thereby deprecating some of the more remote moisture sources. Finally, larger contributions from easterly sources could also be related to circulation differences in terms of a stronger influence by the North Western Pacific Subtropical High in the CTL simulation."

4. L141: The PIN simulations are substantially colder than ERA-Interim?

**Reply:** This sentence was incorrect, the YRV temperature in PIN is close to ERAI. We have rephrased this sentence to talk about CTL only, and moved it further up into the previous paragraph.

5. The CTL time slice used in this study is a 10-year period (L168) or a 5-year period (Table. 1). If the 1996-2005 period is selected, 1997/1998 strong El Niño is selected.

**Reply:** Thank you for this important comment, in fact we did run 10-year periods for all simulations, and we corrected Table 1 accordingly. Indeed, the CTL time slice of the originally submitted manuscript included the 1997/1998 ENSO event. In response to the other reviewer's comments, we made a statistical comparison of the impact of the ENSO event, and the years 2001-2010. As layed out in the text related to Fig. R1 below, we can show that excluding the ENSO event does not substantially impact our results. We explain now our rationale in the revised manuscript, and keep the analysis period to the years 1996-2005.

6. PIN is pre-industrial simulation and CTL is present simulation. Therefore, the comparison between ERAI and CTL is different from that between ERAI and PIN. Therefore, I would suggest the authors reorganize this manuscript solely with results from LGM and PIN simulations. The comparison between CTL and RCP could be formed into another manuscript.

**Reply:** Yes, the pre-industrial (PIN) and control (CTL) simulations are not identical in terms of climate forcing, among other differences. However, noted from the comparison in Fig. 1, the the global and regional

(YRV) temperatures of PIN correspond actually better for the PIN simulation than CTL, with the PIN simulation being in between the CTL and ERAI global mean temperatures. We would like to emphasize that it is a central aspect of our study to include results from more than one climate model in this study, thereby underpinning the robustness of our results from a Lagrangian moisture source diagnostic applied to climate model data. This helps to establish a broader climate system perspective in our study, which would vanish if we split the manuscript in a past and future climate part. In our revisions, we have carefully considered where to further clarify and strengthen our main objectives and results, namely the usefulness of our moisture source and transport perspective to understand hydroclimate across models and climate states.

**Technical Comments**

I would suggest the authors thoroughly check these technical details in this manuscript before submitting the revised manuscript.

Thank you for these detailed technical suggestions, which we will correct during the revision.

1. It is "Yangtze River", not "Yangtse River". There are several "Yangtse River" in the text.

**Reply:** corrected.

2. The usage of "screen-level temperature" and "near-surface temperature". I would suggest using one of "near-surface temperature" or "surface air temperature".

**Reply:** We changed all instances to "surface air temperature".

3. RCP is the abbreviation for "Representative Concentration Pathway", not for "Reference Climate Projection" (caption of Fig. 1).

**Reply:** corrected.

4. I would suggest the authors only use "Celsius (°C)" to describe the temperature and the associated difference.

**Reply:** We prefer to use the Celsius scale for describing temperature, and the SI unit Kelvin for differences, unless the journal formatting guidelines state otherwise.

5. Some expressions should be uniformly used in this study.

5.1 "Western Pacific" instead of "West Pacific".

**Reply:** corrected.

5.2 "South China Sea" instead of "south China Sea". 5.3 "Tibetan Plateau" instead of "Tibetan plateau".

**Reply:** corrected.

6. L288: While the authors say "However, some differences can be seen." The following sentence gives a "similar to ERAI".

**Reply:** This sentence has now been moved further down in the paragraph.

7. Some descriptions are with wrong figure numbers.

7.1 L297: "Fig. 4d" should be "Fig. 4c". Also, "both climate models" should be CTL simulation.

**Reply:** corrected.

7.2 L300: "Fig. 4d" should be "Fig. 4c".

**Reply:** corrected.

7.3 L325: "Fig. 5b and c" should be "Figs. 5c and 5e"?

**Reply:** corrected.

7.4 L372: "Fig. 7b and c" should be "Figs. 7b and 7d"?

**Reply:** corrected.

7.5 L373: "Fig. 7e" should be "Fig. 7c"?

**Reply:** corrected.

7.6 L384: "Fig. 8b" should be "Fig. 8c"?

**Reply:** corrected.

7.7 L388: It seems that Fig. 8d does not indicate "and directly to the east towards Bangladesh".

**Reply:** this sentence has been corrected to "The largest absolute increase can be seen southeast of the YRV towards the Taiwan Strait.".

7.8 L390: "..., coastal regions of the South China Sea and the Bay of Bengal towards India" seems with no significant increase.

**Reply:** This sentence has been corrected to: "…from the Taiwan Strait east towards the ocean regions of the Western Pacific".

8. Figure 7: "RCP" should be "LGM".

**Reply:** corrected.

**References**

Harrison, S.P., Bartlein, P.J., Brewer, S. et al. Climate model benchmarking with glacial and mid-Holocene climates. Clim Dyn 43, 671–688 (2014). https://doi.org/10.1007/s00382-013-1922-6.

**Reviewer#2: Review of WCD-2022-52 by Astrid Fremme et al. 2022**

This manuscript aims to examine how the hydroclimate of the Yangtze River Valley changes along different climates, from the last glacial maximum to the present, and then under a future projection scenario. This fact is based on the comparison of the moisture sources for the target region. The authors used two model outputs from climate models CAM5.1 and NorESM1-M and ran the FLEXPART model trying to obtain the moisture sources for the Yangtze River Valley. ERA-I was also used.

While the exercise of using the different datasets to run de FLEXPART model is novel, there are several major concerns about the modeling, the data used, calculation, comparison, and interpretation that lead to many doubts to this reviewer; many of them related to the robustness of the statistical analysis of this work.
I believe substantial modifications are necessary and thus I would reject this manuscript if no major changes are made.

**Reply:** We are grateful for the reviewer's thorough critical comments. Here we how we have addressed the issues raised by the reviewer in the revised manuscript.

My main comments and reasons for this decision are:

1. My biggest concern is the robustness of the statistics. Throughout the paper, there is no statistical analysis for the comparison of the simulations carried out for the present time (control period). It is needed an annual and seasonal comparison between the dataset used and derived variables from the FLEXPART outputs for the control periods. A visual comparison of the obtained fields, which are shown in the manuscript as maps, is not sufficient to conclude whether the moisture sources are similar or not. A statistical study is necessary. Typical statistics are used in this type of analysis: mean absolute error (MAE), root mean square error (RMSE), Pearson's correlation (R), Bias (B), standard deviation (STD), and coefficient of determination and variation.

**Reply:** We note that several figures already include statistical information to indicate the uncertainty (shaded standard deviations in Fig. 3, and Figs. 5-9). We agree that adding more quantitative, statistical information about the differences between the different results will further strengthen the manuscript, and enhance the visual interpretation of the results. In the revised manuscript, we have now consistently backed up the visual interpretation using the RMSE and Bias as considered useful. To this end, we have added a new Table 2 that collects RMSE and Bias for in particular difference plots for the analysis domain and the YRV. Furthermore, we added the RMSE and Bias throughout the discussion of the seasonality series, such as displayed in Figs. 3, 5, 7 and 9.

**Table 2:** Differences of MJJ precipitation and moisture source contribution between model simulations and time slices for YRV and entire analysis domain expressed in terms of RMSE and Bias (mm day$^{-1}$).

|  | YRV RMSE | YRV Bias | domain RMSE | domain Bias |
|---|---|---|---|---|
| **P CTL-ERAI** | 1.536 | -1.391 | 2.395 | -0.989 |
| **P PIN-ERAI** | 1.861 | -1.340 | 2.917 | -0.220 |
| **E CTL-ERAI** | 0.081 | 0.015 | 0.046 | -0.006 |
| **E PIN-ERAI** | 0.150 | -0.128 | 0.051 | -0.021 |
| **P LGM-PIN** | 0.839 | 0.298 | 2.236 | -0.768 |
| **E LGM-PIN** | 0.034 | 0.030 | 0.020 | 0.001 |
| **P RCP-CTL** | 0.441 | -0.154 | 0.588 | 0.123 |
| **E RCP-CTL** | 0.049 | -0.027 | 0.019 | -0.004 |

2. My other big comment is about the selection of the length of the period to compare. What is the reason for the selection of these 10 years (in the table appears that is 5 years...)? Does decadal variability play some role in the results obtained? The author comment that the period is neutral, but in the selected period as control exits a strong ENSO event, 1997-98, and during the decade the sign of the Pacific Oscillation (POD) was the same, and it is known that the East Asia Monsoon precipitation is affected by ENSO, but the POD strengths this relationship when positive. The Indian Ocean Dipole, which is the role of this mode over the region? If the hydroclimate variability and changes along the time (past-present-future) of a region where the EASM affects is the goal of this paper, it needs to be into account as the other modes commented. The authors should use 30 years as usual for climatological studies.

**Reply:** The reviewer raises a valid point here. First, we need to clarify that the 5 years listed in Table 1 are a mistake that has been corrected in the revised manuscript (initially, before submission, the analysis was only done for a 5-year period). To address the comment by the reviewer, we consider it useful to provide some context on the data availability and the computational requirements of our method. First, we use 10-year time slices from the climate model simulations as basis for running the FLEXPART transport model. After this step, the WaterSip moisture source diagnostic is run based on the FLEXPART output data. Each step of this

post-processing is computationally demanding, both in terms of supercomputer time, and storage, in particular for long time periods.

In the process of extending our initial 5-year analysis time period to a 10 year period, we went backward in time and included the 97/98 ENSO event, as also noted by reviewer #1. An comparison of the entire period 1996-2005 with and without the ENSO 97/98 years shows that the average CTL moisture sources including the ENSO years (Fig. R1a) and without (Fig. R1b) are indistinguishable. The difference between the two panels shows a very small increase in moisture sources to the east of the YRV (Fig. R1e) with an RMSE of 0.020 mm day$^{-1}$ and a bias of 0.011 mm day$^{-1}$ in the YRV, and an RMSE of 0.005 mm day$^{-1}$ and a bias of 0.000 mm day$^{-1}$ in the analysis domain. We also re-ran the moisture source analysis from the CTL simulation for the years up to and including 2010, an analysis which took us more than two month to complete. Again, the average moisture sources for the years 2001-2010 (Fig. R1c) are very similar to the earlier analysis period, with and without ENSO years. The differences are located partly in the YRV, and are in particular related to the years 2009 and 2010, which also was part of an ENSO cycle (RMSE 0.011 mm day$^{-1}$, bias 0.002 mm day$^{-1}$). Interestingly, in comparison to the MJJ-average inter-annual standard deviaion of the 1996-2005 period, excluding the ENSO years appears negligible (Fig. R1d). In the revised manuscript, we now briefly explain our rationale for consistently using the period for the years 1996-2010 for our analysis.

[Figure]

Figure R1: Comparison between MJJ moisture sources (mm day$^{-1}$) for the YRV during years 1996-2005 in simulation CTL with NorESM1-M using (a) all years from 1996-2005, (b) all years except the ENSO years 1997-1998, and (c) all years from 2001-2010, (d) the MJJ-average inter-annual standard deviation, (e) the difference between 1996-2005 and the same period without 1997/1996, and (f) the difference between 2001-2010 and 1996-2005. Note the different color scales in panels (d) and in panels (e, f).

Regarding the suggested expansion of our analysis to a full 30 years, we not that this is currently far beyond the computational resources available to the authors, essentially triplicate the already large computational demands of this analysis method. The large computational demand is a consequence of the offline trajectory calculations. The trajectory model requires 3h or 6h three-dimensional output data on all model levels from a climate model to trace airmass motion with the trajectories. Climate model data at this time resolution are not regularly archived, and thus both rare and extremely large. For example, such data are not part of regular CMIP archiving, and require climate model re-runs to apply. The now included statistical analysis and quantification of differences done in response to comment #1 still allow to draw conclusions with the given limitation to a 10-year analysis period. We now suggest longer analysis period for climate studies in the conclusions.

We agree that a study with the goal to comprehensively investigate the hydroclimate variability along time for the EASM region would need more comments about the role of ENSO and POD in the study region. The main goal of our study is however to investigate if the moisture source perspective can provide insight in

climate model simulations to study hydroclimate changes through time in a key region of the world's climate. We more strongly emphasise this perspective now throughout the manuscript.

3. To resolve these issues, it is necessary to extend the study period as much as possible, compare a longer period, and do statistical analysis.

**Reply:** We have addressed these points in the revised manuscript, as outlined in our answers to point 1 and 2 above.

Overall, a more critical discussion of the limitations and uncertainties of the study seems necessary. Without these questions (methodological and analytical issues) resolved, this reviewer cannot consider this manuscript for publication at this stage.

Other aspects need more attention by the authors:

4. The input data for running FLEXPART model have different characteristics in terms of vertical and horizontal resolution. Is the number of particles modeled the same in the different experiments? If they are the same, the model preconditions are different, and this fact could affect the results and the interpretation of the field comparison. This needs to be checked and explained in the manuscript.

**Reply:** As specified in Sec. 2.2, we continuously release 50'000 particles in the higher-resolution CAM5.1 simulations, and 25'000 particles in NorESM1-M. Since the particles always represent the entire mass in the target area, albeit with a different contribution per trajectory, the main difference beyond a certain, larger number of trajectories will be to provide more spatial detail of the moisture source information. To confirm our expectation that the number of trajectories that we apply does not play a major role for the overall results of the analysis, we have performed a sensitivity run of the moisture source diagnostic with all 50'000 particles from the CAM5.1 simulations, and in comparison one where every other particle is skipped, i.e. using 25'000 particles (Figure R2). The RMSE in the YRV region is 0.027 mm day$^{-1}$, and the bias 0.025mm day$^{-1}$. For the entire displayed analysis domain, the rmse is 0.006 mm day$^{-1}$ and the bias 0.000 mm day$^{-1}$. These differences are about a factor of 10 smaller than the differences we see between PIN and CTL. We state in the revised manuscript that we have performed a sensitivity test regarding the number of particles, which only showed minor impact on the moisture source contributions.

[Figure]

**Figure R2:** Comparison between annual mean moisture sources (mm day$^{-1}$) for the YRV during year 2002 in simulation PIN with CAM5.1 using (a) 50'000 particles, (b) 25'000 particles, and (c) difference.

5. The authors comment that the thresholds in WaterSip were changed, and the selected RH% , for instance, was determined for NorESM dataset. Why not other data or reanalysis? Could this affect the outputs of the model when used for the future? The diagnostic for the imposed thresholds was done only for one year! Which year? Why this one? If the year selected was another, could the threshold change? ... How this affect the results?

**Reply:** The thresholds were determined originally (and extensively) for the reanalysis dataset (see e.g., Sodemann and Stohl, 2010), and here we essentially cross-check that the same thresholds are meaningful to apply with the climate model data. The sensitivity to the RH threshold, as detailed in Appendix A1 and A2 were tested for the year 2002 of the CTL simulation, as stated in the figure captions and now also in the text. Testing the threshold sensitivity on the basis of all 25'000, continuously released trajectories for each 3h

time-step of year 2002 does provide a sufficient basis to robustly evaluate the threshold sensitivity. It is not obvious to us how and why the impact of this parameter could change more substantially by year than the already large sensitivity displayed in Fig. A2. In the end, a lower threshold will lead to a larger total precipiation. However, with the focus of the offline diagnostic being on the moisture source regions for the Lagrangian precipitation estimate $\Pi$ (Sec. 3.2), the location and relative contribution of the moisture source regions themselves are hardly sensitive to the RH threshold. We have now added a paragraph to the conclusions to emphasize these points:

"Being a first application of this moisture source diagnostic to climate model data rather than reanalysis data, we see that the same thresholds of the moisture source diagnostic of Sodemann et al. (2008) work for a dataset without increments in humidity from data assimilation. We note an on average larger faction of precipitation that is accounted for in terms of moisture sources, potentially as a potential consequence of the absence of data assimilation."

6. Only one emission scenario, RCP6.0, was used in the study. Why? RCP6.0 represents an intermediate scenario, and it is not very used in climate change studies. What about RCP8.5?

**Reply:** As mentioned in our reply to comment #2, the offline moisture source diagnostic is a computationally very demanding diagnostic method, that requires specific output for running the trajectory model. Such output was made available specifically from within the EVA e-science project, and only a RCP6.0 scenario has been run, and is currently available. A re-run with a different, higher-emission climate scenario would be interesting, but is beyond, and we argue also not necessary for the claims made in this pilot study of moisture source diagnostics with climate model output. To stress this point, we now state in the conclusions: "It would be interesting to follow up in the future with a higher emission climate scenario, to reveal potentially stronger responses in the hydroclimate."

Many of the comparisons in the manuscript, such as those derived from figure 3, etc..., were in absolute values. 1.0 mm/day seems a small quantity, but it is not small if the variables to compare have a range from 4 to 8mm/day. Better in %.

**Reply:** We checked that the units are meaningful and interpretable.

There are many typo errors in the manuscript (table, figures, captions, ...). Please check the manuscript carefully when the revision is done.

**Reply:** We carefully checked the revised manuscript for typos, in particularly in the figure and table captions, and harmonized to British English spelling.

This review has further comments on the results and their interpretation, but these are not necessary at this stage of the review until the above is resolved.

---

## Author Response (AR2)

Dear Martin Singh,

we have now completed the minor revisions in response to the reviewer comments and your editorial comments. We feel we could address all comments adequately. In the responses below, we add our replies in blue text, starting with the keyword "Reply". We also made numerous small changes throughout the manuscript to correct grammatical and spelling errors, and to facilitate the readability of the manuscript. All changes are apparent from the track changes version of our revised manuscript.

With best regards, on behalf of all authors,
Harald Sodemann

**Reviewer comments:** Thanks for the review. The authors have responded to most of my comments, but as I had already commented in my first review it is necessary to increase the study period and use another climate change scenario. These two comments are the main problems of the article from the point of view of this reviewer.
The first point is related to the 10-year period, any climatology must be longer than this period, and more if compared with present and past periods.
The second, about the RCP6.0 scenario, is due to the fact that it is not a usual scenario, since it is a medium warming scenario. The fact that it is the one that researchers have from a previous project does not justify that it is a usual scenario for research articles in which the authors want to show changes in the future times.

These two aspects should be corrected in the opinion of this reviewer.

**Reply:** we thank the reviewer for their time to read our revised manuscript. We provide answers to these two comments below.

In any case, and if in view of the comments of the other reviewer, the editors consider that it should be taken into consideration for publication, there are more comments that should be addressed:

1- The title must be modified to show better the content of this paper:
a) the analysis is a pilot study, and this should be paper in the title
b) the study does not show future climates, it is only one, and a non-usual one.
c) the paper shows an analysis on the Yangtze River region, not for the East Asian Summer Monsoon.
Monsoon dynamics are more complicated than the analysis presented here. This should be corrected throughout the paper. Evidently, the region of the Yangtze River is affected by the Monsoon, and this can continue to be indicated in the introduction and in the discussion, as well, as the relationship with the different modes of variability, but not indicate that the summer monsoon is studied.

**Reply:** We have now modified the title in response to the reviewer's comments as "Model-simulated hydroclimate *in* the East Asian Summer Monsoon *region* during past and future *climate*: *a pilot study with a moisture source perspective*". In the revised title, we mention the pilot character (comment 1a), changed the climate to singular form (comment 1b), and use a more loose formulation for the location of the study (comment 1c). We think this is justified, since the YRV precipitation processes reach far into East Asia. We mention in the first sentence of the abstract and early in the introduction that the study is focused on the YRV, and thus deem our title to be appropriate for attracting a wider readership, while not misleading readers about the specific focus location of our study.

2- If the 10-year period is finally accepted, it should be explained as a weakness of the study.

**Reply:** We have now at numerous locations throughout the manuscript (Abstract, Method, Results, Discussion and Conclusions, see track changes version of the manuscript) clearly flagged the 10-year time period as a limitation of the study, and point to the importance of the conventional 30-year periods to dampen the impact of climate variability.

3- And if the climate change scenario is also maintained, its characteristics and the reason for this choice must be scrupulously explained, and that this work should have been repeated under other scenarios

**Reply:** We understand that the RCP6 scenario is less commonly analyzed in studies than the RCP8.5 scenario. However, we also would like to point out that the RCP6 is a legitimate tier 1 scenario of the Climate Model Intercomparison Project simulations (Taylor et al., 2012). We therefore disagree that it should be necessary to repeat the study with other scenarios. We now state the characteristics of the RCP6 scenario in the methods, and point out at several locations throughout that a higher-emission scenario could be used to potentially obtain more emphasized responses in the moisture source changes.